# Prediction of glycopeptide fragment mass spectra by deep learning

Yi Yang [1] ✉ & Qun Fang [1,2] ✉

Deep learning has achieved a notable success in mass spectrometry-based proteomics and is now emerging in glycoproteomics. While various deep learning models can predict fragment mass spectra of peptides with good accuracy, they cannot cope with the non-linear glycan structure in an intact glycopeptide. Herein, we present DeepGlyco, a deep learning-based approach for the prediction of fragment spectra of intact glycopeptides. Our model adopts tree-structured long-short term memory networks to process the glycan moiety and a graph neural network architecture to incorporate potential fragmentation pathways of a specific glycan structure. This feature is beneficial to model explainability and differentiation ability of glycan structural isomers. We further demonstrate that predicted spectral libraries can be used for data-independent acquisition glycoproteomics as a supplement for library completeness. We expect that this work will provide a valuable deep learning resource for glycoproteomics.

Liquid chromatography coupled with tandem mass spectrometry (LC-MS/MS) is the method of choice widely used in proteomics[1] and glycoproteomics[2,3]. At the heart of proteomics data analysis is peptide identification by matching fragment spectra to theoretical or experimental spectra for candidate peptides[4]. Most commonly used proteomics[5–8] or glycoproteomics[9–16] search engines are based on database searching, where peptide spectrum matches (PSMs) or glycopeptide spectrum matches (GPSMs) are scored on the presence of fragment ions theoretically generated from peptide sequence and glycan but largely disregard fragment ion intensities. As a complementary approach, spectral library searching correlates the intensity pattern of fragment ions of the analyte to library spectra typically constructed from previous identification data[17], which has been reported to yield more discriminative match scores than database searching for data-dependent acquisition (DDA) analysis[18–21]. Spectral libraries are also commonly used for the analysis of data-independent acquisition (DIA) experiments[22], achieving deep proteome coverage with quantitative consistency in conventional proteomics[23–25], phosphoproteomics[26,27], ubiquitin proteomics[28], and glycoproteomics[29,30]. However, the incompleteness of library coverage determines the upper limit of the identification capacity by spectral library searching. In addition to experimentally recorded spectra library, computational methods for the prediction of peptide spectral libraries are of growing attention.

Over the years, machine learning and in particular deep learning approaches have become increasingly prevalent and beneficial in proteomics[31–33]. Efforts have been made using deep neural networks for the prediction of peptide properties and behaviors throughout the MS-based proteomics workflow, including detectability related to digestibility by proteases[34,35], retention times in LC[36–38], collisional cross sections in ion mobility spectrometry[39], and fragment ion intensities in MS/MS[40]. Deep learning-based methods for predicting the peptide fragment intensities include pDeep[41–43], DeepMass:Prism[44], Prosit[45], AlphaPeptDeep[46], and many subsequent ones now represent the state-of-the-art for various tasks. Fragment spectrum prediction has been used for improving DDA-based peptide identification by integrating the intensity information into PSM scoring, resulting in better sensitivity and specificity[47–49]. DIA data analysis has also benefited from peptide fragment spectrum prediction[50,51]. Predicted spectral libraries can be generated directly from protein sequence databases[51]. Proteome-wide spectral library prediction, coupled with deep learning-based feature scoring models have been developed to discriminate real signals from noise, has enabled extraordinarily deep proteome coverage in high throughput DIA analysis without the need

[1]ZJU-Hangzhou Global Scientific and Technological Innovation Center, Zhejiang University, Hangzhou 311200, China. [2]Department of Chemistry, Zhejiang University, Hangzhou 310058, China. ✉e-mail: y_yi@zju.edu.cn; fangqun@zju.edu.cn

of experimental spectral libraries[52]. Deep learning models were further specialized for specific post-translational modifications (PTMs), such as DeepPhospho[53] for DIA phosphoproteomics and DeepFLR[54] for phosphorylation site localization. Nevertheless, yet current methods fail to predict fragment spectra of intact glycopeptides.

In contrast to deglycosylated peptides, intact glycopeptides maintain the peptide-glycan link and therefore can provide information on the peptide sequence, linked glycan structures, and glycosite[55]. The glycan moiety is an elaborate structure composited of different monosaccharides and variable linkages among them. The existing tools for peptide property prediction mostly use long-short term memory (LSTM)[41–44,51], gated recurrent unit[45], or transformer-based models[53,54]. These models can only process linear input of peptide sequences (with simple PTMs considered as indivisible tags), whereas they do not cope with the glycan structure. Moreover, fragmentation behaviors of intact glycopeptides in MS/MS differ from non-glycosylated peptides. Higher-energy collisional dissociation (HCD) with stepped collision energy (CE), the most common fragmentation strategy for N-glycopeptides, provides sequential cleavages of both the glycan and peptide bonds[56]. This results in merged spectra containing not only the peptide fragments (b/y ions) but glycan fragments (B/Y ions), which are not covered by the existing models for peptide fragment spectrum prediction. Novel architectures, like graph neural networks that were recently adopted for de novo sequencing of glycans[57], are required to learn glycan structures and their relevance to the fragment ions.

Herein, we present a deep learning-based framework called DeepGlyco for the prediction of MS/MS spectra of intact glycopeptides. While the input peptide sequences are processed by conventional LSTM networks, the glycan structures are resolved by introducing the tree LSTM networks. Putative fragmentation pathways of structure-specific glycans are modeled by graph neural networks with the attention mechanism, enabling the explanation of possible origins of the predicted fragment ions. This feature is beneficial to differentiating glycan structural isomers. We further demonstrate that predicted spectral libraries are also suitable for analyzing DIA data of glycopeptides as a supplement for library completeness. We expect that this work will provide a valuable deep learning resource for glycoproteomics.

## Results

### Model design for intact glycopeptide MS/MS spectrum prediction

DeepGlyco inherits the LSTM-based model architectures for peptide property prediction[42,46], while it is extended with additional modules to predict glycan fragment intensities (Fig. 1). A glycopeptide input is split into peptide and glycan moieties before being fed into the model. The peptide moiety includes the amino acid sequence and modifications, which is represented by one-hot indicators and element compositions, respectively, and then processed by a linear LSTM network similar to the models for peptide MS/MS prediction. The glycan moiety is a tree with the one-hot-encoded monosaccharides as the nodes and their linkages as the edges. A tree LSTM network traverses the glycan tree in the bottom-up direction (from the non-reducing end to the reducing end of glycan). The tree LSTM is a generalization of the standard LSTM that adapts to tree-structured network topologies by combining the hidden states of all sub-trees[58]. In this study, a sub-node summing variant of tree LSTM ignoring the order of glycan branches

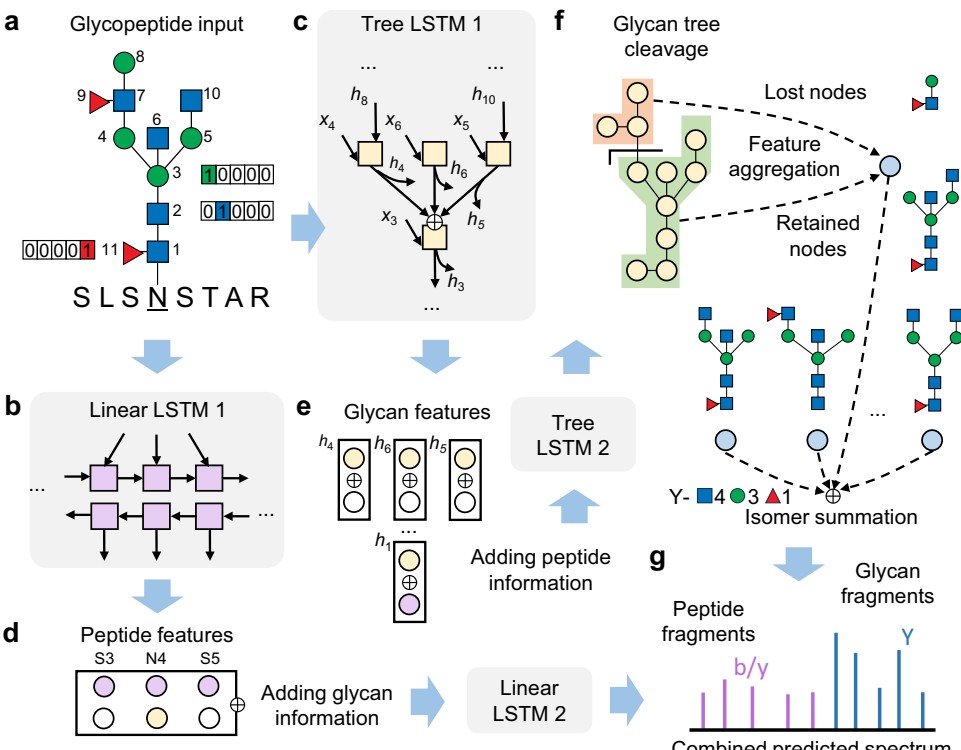

**Fig. 1 | Overview of the deep learning model for glycopeptide fragment spectrum prediction.** **a** The input glycopeptide comprises a peptide sequence and a glycan tree, where monosaccharides are one-hot encoded. **b** The peptide sequence is processed by a linear LSTM network. **c** The glycan tree is traversed by a tree LSTM network. **d**, **e** The peptide features extracted by the linear and the glycan features by the tree LSTM are fused with each other. Then peptide features are processed by another linear LSTM network to predict the relative intensities of peptide b/y fragments. The glycan features are traversed by another tree LSTM

network, updating the feature of each monosaccharide node in the glycan tree. **f** Features of potential cleavage sites are aggregated from the monosaccharide nodes that are lost or retained after the cleavage. Features of structure-specific glycan fragments are aggregated from the corresponding cleavages to predict the relative intensities of Y ions, where structural isomeric fragments are combined. **g** The peptide and glycan fragment ions are finally merged to form the output glycopeptide spectrum. The monosaccharide symbols are defined in Supplementary Table 1.

was employed because the order information is hardly provided by the conventional HCD MS/MS-based glycopeptide identification method. Then the information of peptide and glycan is interchanged by a fusion of the features extracted by the linear LSTM and the tree LSTM. The precursor charge state is also concatenated to the peptide and glycan features. The peptide features are processed by another linear LSTM network to predict the relative intensities of peptide b/y fragments, yielding the peptide part of the output MS/MS spectrum.

The glycan features are updated by another tree LSTM that traverses the glycan tree in the top-down direction. Thus, features of each node in the glycan tree contains the information of monosaccharides downward, upward, and in other branches. Each potential cleavage site splits the glycan tree into nodes at the reducing end (lost after cleavage) and the non-reducing end (retained in the Y fragment ions). Features of the lost and retained nodes are aggregated with an attention mechanism, turning into the feature vector of the cleavage (Fig. 1f and Supplementary Fig. 1). Then the relative intensity of each structure-specific Y fragment is computed by feature aggregation of corresponding cleavages. Fragment isomers with the same monosaccharide composition are combined by summing up their intensities, yielding the glycan part of the output MS/MS spectrum. The updated peptide features and glycan features are also aggregated to a vector and then transformed into a scalar value to predict the ratio of the peptide fragment intensity to the whole MS/MS spectrum. Finally, the peptide and glycan fragments are merged by the ratio, forming the output glycopeptide spectrum. The model contains ~7 million parameters in total (including ~5 million for the peptide part and ~2 million for the glycan part). Details of the model architecture are described in the Methods section.

Specially for complex and hybrid type glycans, we further constructed a spectrum prediction model that incorporates with fragment ions from the glycan branches at the non-reducing end (Supplementary Fig. 2). These ions, referred to as B ions in this study, include the whole branch falling off the glycan core, fragments originated from cleavages within the branch, as well as the branch and its fragments with the adjacent core mannose. Analogous to the Y ions, the relative intensities of B ions were predicted by feature aggregation of corresponding branch cleavages and then merged to the output glycopeptide spectrum. Oligomannose structures in the high-mannose or hybrid glycans were not taken into consideration due to lack of B ions with multiple mannoses in glycopeptide MS/MS spectra[13].

## Performance evaluation of glycopeptide MS/MS spectrum prediction

We trained and validated the model with datasets of diverse organisms acquired on Orbitrap mass spectrometers with distinct instrument settings[10,13,15,30,59-62] (Supplementary Table 2 and 3). In each dataset, redundant spectra were removed by combining them into consensus spectra[30] (one spectrum per glycopeptide precursor). Before a dataset was used for training, it was randomly partitioned into three subsets, where 3/5 were used for fitting the model parameters, 1/5 to control for overfitting, and the remaining 1/5 not involved in training (holdout) for performance estimation. The spectral angle loss[45] (SA) was used as the objective function for spectrum prediction because of its higher sensitivity than dot product (DP). SA values of the peptide part, the glycan part, and the whole spectrum, as well as prediction error of the fragment intensity ratio, served as four objectives for simultaneous optimization and thus the model was trained by multi-task learning.

Benchmarking on the Mouse 1 and Human 1 datasets, fragment spectrum prediction achieved very high similarities (Fig. 2a). The median SA values of the peptide part were 0.22–0.16 (corresponding to DP of 0.94–0.96), those of the glycan part were <0.11 (DP > 0.98), and those of the whole spectrum were <0.16 (DP > 0.97) for the holdout set. No substantial discrepancy of metrics was observed between the training and holdout sets, demonstrating that the model was not

strongly overfitted. In addition to the consensus spectra, we evaluated the model stability over replicate spectra, revealing a larger deviation in the similarity distribution (Supplementary Fig. 3). Example spectra are shown in Supplementary Data 1, showing the variation in glycopeptide fragmentation among replicates. Due to the limitation that our model ignores the order of glycan branches, isomeric glycopeptides may be one of the factors contributing to the variation among replicated spectra. The trained model was also tested across different instrument settings (Supplementary Figs. 4–7). Trained with Mouse 1 and tested on the other mouse datasets (Mouse 2–4), the median SA values of the peptide part were 0.28–0.26 (DP of 0.90–0.91), those of the glycan part were 0.22–0.18 (DP of 0.94–0.96), and those of the whole spectrum were 0.24–0.21 (DP of 0.93–0.95). Trained with Human 1 and tested on the other human datasets (Human 2–4), prediction of the peptide part was less accurate (SA of 0.38–0.32, DP of 0.82–0.88). The results indicated that peptide part was more susceptible to CE settings. This is reasonable since in a typical HCD spectrum of glycopeptide with stepped CE, peptide fragments are usually considerably less intense than glycan fragments (Fig. 2b). Within each dataset, the intensity ratio between peptide and glycan fragments varied in a wide range across different glycopeptides, while most glycopeptides had lower intensity ratios (Supplementary Fig. 8). Variations of the ratio were also observed across datasets, probably associated with the experimental condition. Prediction errors of the intensity ratio are visualized in Supplementary Fig. 9 and 10, and its effect on the overall spectral similarity was investigated (Supplementary Note 1, as well as Supplementary Figs. 11 and 12). The impact of incorrectly identified spectra in the training data was also explored (Supplementary Note 2, as well as Supplementary Figs. 13 and 14).

Since the change of instrument settings gave an impact on the prediction accuracy, we further finetuned the pre-trained models on datasets with other CE settings (Fig. 2c, d, Supplementary Figs. 15–18). The model trained with Mouse 1 was fine-tuned with Mouse 2, the median SA of peptide part on Mouse 3 (with the same CE setting as Mouse 2) was improved from 0.26 – 0.21 (corresponding to DP from 0.91 to 0.94). The results were similar on human datasets, the median SA of peptide part on Human 4 was improved from 0.39 – 0.24 (DP from 0.82 to 0.93) after finetuning. For cross-organism prediction (Supplementary Fig. 4b, c, training with Mouse 1, testing on Human 2, Human 4, and Yeast; Supplementary Fig. 15a, fine-tuning with Mouse 2, testing on Human 1 and Human 3; Supplementary Fig. 6b, training with Human 1, testing on Mouse 2 and Mouse 3; Supplementary Fig. 15b, fine-tuning with Human 2, testing on Mouse 1, Mouse 4, and Yeast), the median SA values of the peptide part were 0.34 ± 0.02 (corresponding to DP of 0.86 ± 0.02, mean ± standard deviation, similarly hereinafter), those of the glycan part were 0.23 ± 0.07 (DP of 0.93 ± 0.04), and those of the whole spectrum were 0.27 ± 0.07 (DP of 0.89 ± 0.04). In contrast to model training/finetuning on individual datasets, we also explored the effect of combining multiple datasets to create a larger dataset for model training (Supplementary Note 3 and Supplementary Figs. 19–22). The results suggest that incorporating more datasets of different instruments or organisms for model training would lead to better generalization.

The model with branch fragments was retrained and validated with the datasets excluding high-mannose glycopeptides. The neural network layers for prediction b/y and Y ions were migrated from the models without B ions and frozen during training. As the intensities of B ions with mono- or disaccharides (oxonium ions) are much higher than other fragment ions, similarity metrics of prediction were computed not only covering the complete spectrum but also for B ions and Y ions separately (Supplementary Figs. 23–30). With the good prediction performance of Y ions staying, B ions achieved quite high similarities (median SA of 0.16–0.06, corresponding to DP of 0.97–0.99, Supplementary Figs. 25 and 27) across different organisms and instrument settings, giving rise to accurate prediction of the whole

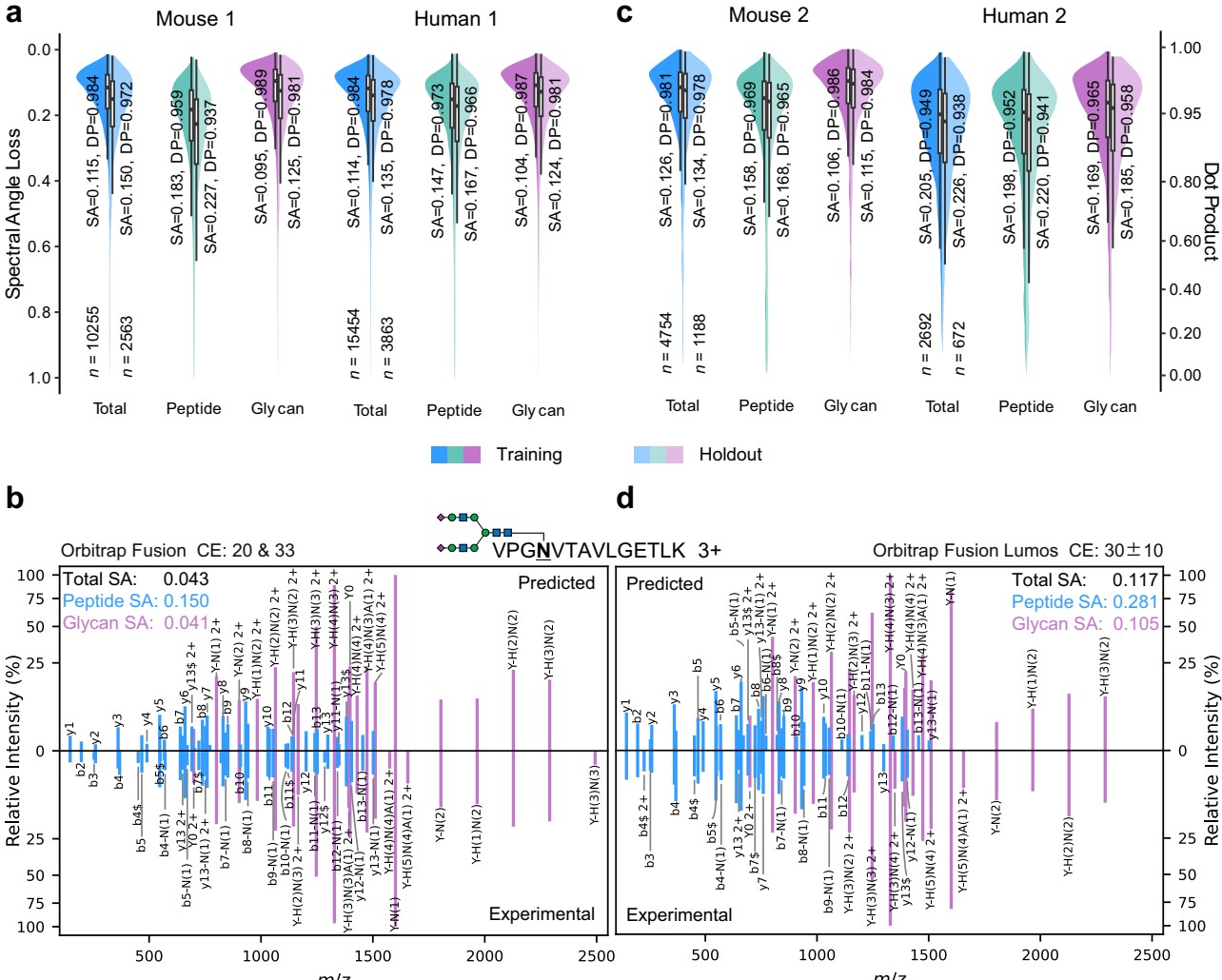

**Fig. 2 | Performance of glycopeptide fragment spectrum prediction.**
**a** Distributions of spectral similarities between predicted and experimental frag-
ment ion intensities for glycopeptides contained in the training or holdout set of
Mouse1 and Human1. **b** Mirror plot of a glycopeptide spectral match comparing
predicted fragment intensities to experimental fragment intensities in Human1.
**c** Distributions of spectral similarities for Mouse2 and Human2 after model fine-
tuning. **d** Mirror plot comparing predicted and experimental fragment intensities in

Human2. Spectral similarities are computed for peptide b/y ions and glycan *Y* ions
separately, as well as for the total spectrum of peptide and glycan ions. In (**a**) and (**c**)
the median values of spectral angle loss (SA) and dot product (DP), as well as data
size (*n*), are indicated. The lower/upper hinges of the boxes indicate the first/third
quartiles, and the lower/upper whiskers extend from the hinges to the smallest/
largest value no further than 1.5 times the interquartile range. Source data are
provided as a Source Data file.

spectrum (median SA of 0.20–0.10, corresponding to DP of
0.95–0.99). Notably, similarity metrics were reported against the
annotated b/y and B/Y ions accounting for 26–34% intensity of the raw
experimental spectra (Supplementary Fig. 31). Other peaks, such as
those sourced from noncanonical fragments or noise signals, were not
covered in this study.

### Differentiating MS/MS spectra of structural isomeric glycopeptides

Based on the prediction model, we explored the potential divergence
of MS/MS spectra of glycopeptide structural isomers with identical
peptide sequences and monosaccharide compositions. The spectral
matches of non-high-mannose glycopeptides were selected from the
MS/MS datasets as query spectra for spectral library searching. For
each spectral match, candidate glycopeptides were generated by
replacing the original glycan with its structural isomers in a predefined
glycan space. The query spectrum was then compared with the pre-
dicted spectrum of each candidate glycopeptide and the similarity
metrics were calculated between them (Fig. 3a). The original

glycopeptide annotations were obtained from StrucGP[13] search results.
Despite different search spaces and scoring mechanisms, the original
StrucGP annotations served as putative correct answer due to lack of
ground truth data.

The ability of predicted spectral library searching to differentiate
glycopeptide structural isomers was evaluated on the holdout datasets
of Mouse 1 and Human 1, as well as a dataset of standard
glycoproteins[13] (Supplementary Table 4). The candidates for each
query spectrum were ranked by a combined similarity score of Y and B
ions (described in the Methods section). We assessed the rescored
results at three levels, i.e., recognition of core fucosylation or bisecting
HexNAc, as well as the correct identity including the former two and all
the branches in a glycan (Supplementary Data 2–4, Supplementary
Figs. 32–37). It should be noted that the branching order is ignored
when describing a glycan structure. The percentage of spectra in which
the correct identity was ranked as the first, second or third candidate
were calculated out of the number of cases with more than 1, 2 or 3
candidates in total (Fig. 3b, Supplementary Figs. 32a and 34a). While
71%–80% of the spectral matches were correct, spectrum prediction

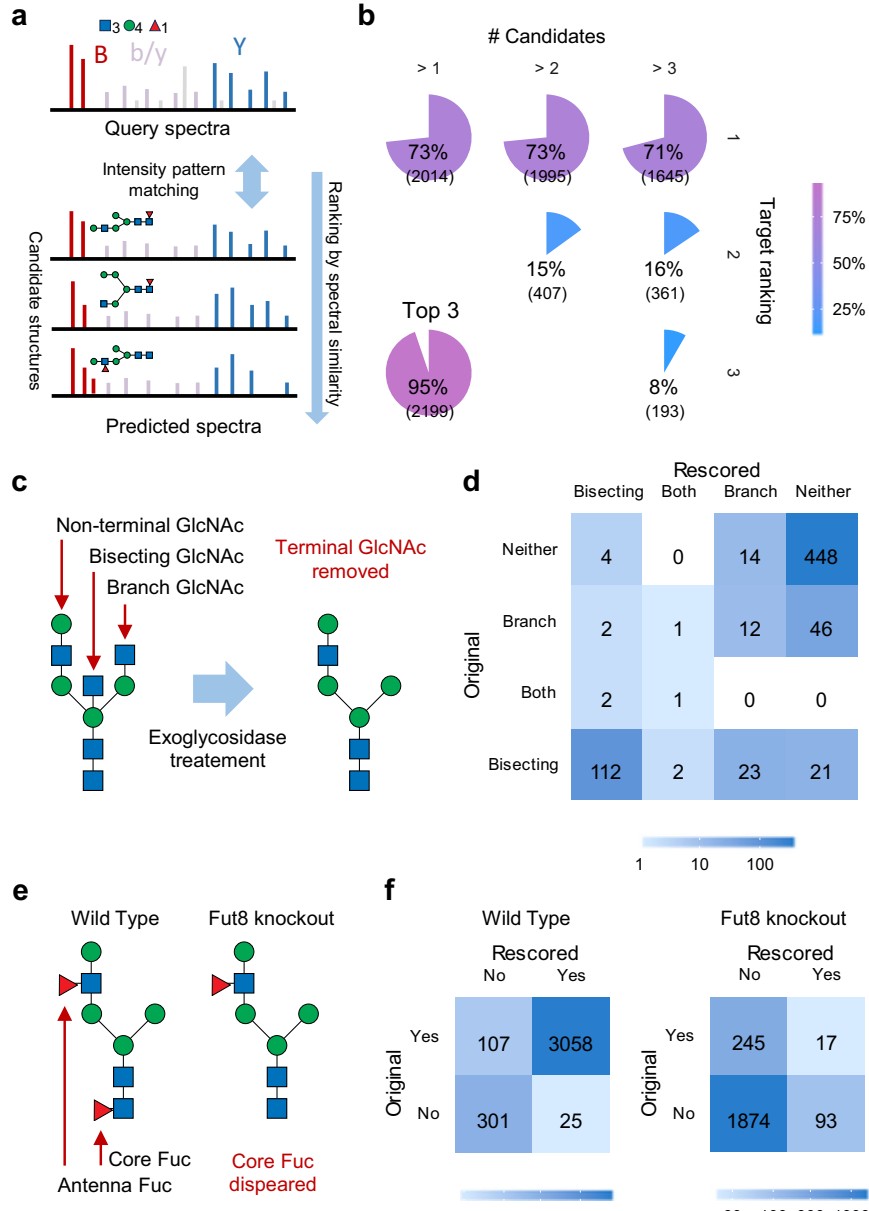

**Fig. 3 | Differentiating structural isomeric glycopeptides using predicted spectral libraries. a** The query spectrum is compared with predicted spectra of candidate glycopeptides with isomeric glycan structures, which are then ranked by the spectral similarity scores. **b** Candidate ranking results of a standard glycopeptide dataset. The percentage of spectra in which the correct identity was ranked as the first, second or third candidate are calculated out of the number of cases with more than 1, 2 or 3 candidates in total. The top-three chart shows the percentage of cases in which the correct identity is ranked among the top three candidates of the total of cases with more than three candidates. **c** Schematic illustration of endoglycosidase treatment for terminal HexNAc removal. **d** Confusion matrix of terminal HexNAc recognition resulted from reanalyzing a dataset of mouse brain with endoglycosidase treatment using a predicted spectral library. **e** Core fucosylation disappears in Fut8 knockout mouses while retains in wild type ones. **f** Confusion matrices of core fucosylation recognition resulted from reanalyzing a dataset of Fut8 knockout and wild type mouse brains. GPSMs with all the candidates belonging to the same category are excluded when calculating the confusion matrices.

enabled ranking the correct identity among the top three candidates in 92%–95% of the cases. Considering the top one candidate for each query spectrum, bisecting HexNAc recognition achieved an accuracy of 88%–95% (Supplementary Figs. 32c, 34c and 36b). Among the positive cases after rescoring, 69%–88% (precision) had been originally annotated as bisecting HexNAc-containing cases, covering 68%–87% (recall) of the original annotations with bisecting HexNAc in total.

The conflicting results of spectral library searching with the original StrucGP annotations were counted according to the mis-identified branch types (Supplementary Figs. 33, 35 and 37, as well as Supplementary Note 4). Among them, confusion between HexNAc-Hex and HexNAc branches accounted for the largest proportion. We

then reanalyzed a dataset of mouse brain where terminal or bisected HexNAc were expected to be removed by exoglycosidase treatments[13] (Fig. 3c, d, Supplementary Fig. 38, Supplementary Data 5). Most of the cases (96%, 448/466) originally annotated as neither bisecting nor terminal HexNAc were retained after rescoring, while 13% (21/158) of the original bisecting HexNAc cases and 75% (46/61) of the original terminal HexNAc cases were reported as dual negatives. Notably, it is still difficult to confirm whether the remaining bisecting or terminal HexNAc after exoglycosidase treatments were false annotation or actual results caused by inadequate reactions. Nevertheless, we present representative cases to interpret the difference of the results by spectral library searching from the original StrucGP annotations.

For a spectral match of IGSYN[H(4)N(3)F(1)]GTAGDSLSYHQGR$^{3+}$, the original glycan structure annotation by StrucGP has a core fucose and a branch terminal HexNAc, whereas a candidate glycan structure with a HexNAc-Hex branch was ranked first after rescoring by spectral library searching (Supplementary Fig. 39). Comparing the query spectrum with predicted spectra of candidate isomers, the core fucose was confirmed by high intensity of the ion Y-H(3)N(2)F(1). However, the original glycan should produce the ions Y-H(4)N(2)F(1), Y-H(2)N(3)F(1) and Y-H(3)N(3)F(1), which were absent or matched with very weak peaks in the query spectrum. The B ion H(1)N(1) was expected to exhibit a higher relative intensity in MS/MS of the rescored glycan structure than the original one. The presence of B ion H(2)N(1) in the query spectrum was also a plus for the rescored glycan structure. Therefore, the intensity pattern of the query spectrum was more similar to that of the rescored structure. We also visualized the glycan fragmentation graph in the model to find the source of these ions (Supplementary Fig. 40). The nodes of fragment compositions and fragment structures are linked by edges with weights reflecting the proportion of fragment isomers. Each fragment isomer is connected with its originating cleavages, where the attention weights can be used to discover the most important cleavages to a fragment isomer. For the original structure, the ion Y-H(4)N(2)F(1) was unique to the cleavage of the terminal HexNAc, and the ions Y-H(3)N(3)F(1) and Y-H(2)N(3)F(1) were generated from the cleavages in the consecutive hexoses. For the rescored structure, the ion Y-H(2)N(3)F(1) could be generated in theory from the simultaneous cleavages of the terminal hexose in the branch (VI) and a core hexose (VII). The attention weights indicated that the cleavage VII was the determinative step. These Y ions were scarcely in the predicted spectrum, which is in accordance with the fact that cleavages are less liable within the core than the branch under relative low collision energy. This also explained the cause of different relative intensity of B ion H(1)N(1) between the two candidate structures. More examples are present in Supplementary Note 5, Supplementary Figs. 41–44.

For the Mouse1 holdout, Human1 holdout, and standard glycoprotein datasets, core fucose recognition achieved an accuracy of 93%–97%, precision of 94%–99%, and recall of 96%–98% (Supplementary Figs. 32b, 34b and 36a). We further reanalyzed a dataset of wild type and Fut8 knockout mouse brain[59] (Fig. 3e, f, Supplementary Fig. 45, Supplementary Data 6). In the knockout samples, all fucosylated glycopeptides should be antenna fucosylated since the specific glycotransferase of core fucosylation had been knocked out. From the wild type samples, 97% (3058/3165) of the cases originally annotated as core fucosylation and 92% (301/326) of negative cases were retained after rescoring, indicating that spectral library searching did not side with negative cases. From the knockout samples, core fucosylated cases reduced from 12% (262/2229) to 5% (110/2229) after rescoring. The majority of the remaining core fucosylated GPSMs were likely false identifications. The challenge of accurate glycan structure identification has not been fully resolved since the estimated false discovery rate was supposed to be 1%. Nevertheless, the results still indicate improvement with the spectrum prediction-based rescoring.

Representative spectral matches are presented in Supplementary Note 6, and Supplementary Figs. 46–50. In some cases, spectral library searching confirmed the original glycan structure annotation by StrucGP with the high intensity of characteristic Y ions of core fucosylation. In others, different candidate structures were ranked first based on the intensity pattern of Y ions and B ions comprehensively. The alteration of peak intensity for different candidate structures could be explained using fragmentation graph in the model.

### Predicting glycopeptide spectral libraries for DIA data analysis
In silico peptide spectral libraries have been proven compatible with DIA analysis and can supplement or sometimes even substitute for experimental libraries[51,52]. For this reason, we explored the ability of predict glycopeptide spectral libraries for DIA analysis. In addition to the fragment spectra, we trained models to predict indexed retention time (iRT)[63] values of glycopeptides (Supplementary Note 7 and Supplementary Fig. 51). On the Mouse 1 and Human 1 datasets, the models achieved high correlation between predicted and observed iRT values (Pearson correlation coefficient $r > 0.97$) for the holdout sets.

A predicted spectral library should contain two levels of information: (1) which glycopeptides should be measured in a sample; (2) their fragment ions and retention time values. Both of them will affect the DIA analysis results. Therefore, we first evaluated the quality of predicted values by keeping the coverage of predicted library equivalent to the experimental library. We benchmarked predicted spectral libraries against sample-specific experimental spectral libraries using DIA datasets in our previous study[30] (Supplementary Table 5 and 6). For evaluation of a fission yeast dataset, we first predicted MS/MS spectra for glycopeptides contained in the experimental library (DDALib), generating a spectral library (PredMS2) using the predicted MS/MS spectra and the original iRT values in DDALib. We also built a spectral library (PredLib) in which both MS/MS spectra and iRT values are predicted. DIA data analysis was performed using GproDIA[30], where statistical control was conducted on both the peptide and glycan parts using a target-decoy approach. The numbers of detected glycopeptides resulting from the predicted libraries were compared to the experimental library (Supplementary Note 8, Supplementary Figs. 52 and 53, Supplementary Data 7). Compared to DDALib, the predicted libraries resulted in a loss of up to 10% detected glycopeptide precursors and site-specific glycans, but a slightly better data completeness. It should be noted that the glycans in the reported glycopeptide precursors were identified as monosaccharide compositions since structural isomers are indistinguishable by current DIA glycopeptide analysis workflow[30]. The term "site-specific glycan" is referred to a glycan composition on a protein glycosite, which contains a group of glycopeptide variants resulting from missed cleavages in protein digestion.

We repeated the above analysis for a human serum dataset using a finetuned model (Fig. 4a–d, Supplementary Figs. 54 and 55, Supplementary Data 8). As serum samples were much more complex than yeast, the glycoform inference option was enabled in DIA data analysis to resolve interference from potential co-eluted and co-fragmented glycopeptides. In average of 3 technical replicate runs, 859 ± 3 precursors of 594 ±2 site-specific glycans were detected using PredMS2 and 798 ± 3 precursors of 553 ± 1 site-specific glycans were detected using PredLib, compared to 799 ± 14 precursors of 539 ± 2 site-specific glycans when using DDALib. Accumulating the 3 replicate runs, 956 precursors of 647 site-specific glycans were detected totally using PredMS2, among which 76% (729) precursors and 81% (522) site-specific glycans were shared in all the replicates. PredLib resulted in 75% (667/893) precursors and 79% (480/605) site-specific glycans shared in all the replicates, compared to 75% (671/892) precursors and 77% (463/599) site-specific glycans when using DDALib, indicating a close data completeness. Considering identifications shared in >50% replicate runs, using the predicted fragment spectra resulted in a gain of 7% (893 compared to 835) precursors and 10% (613 compared to 556) site-specific glycans. Replacing the retention time values led to a loss of 6% precursors (379 compared to 401) and 7% site-specific glycans (833 compared to 893). The library with predicted spectrum performed better than the experimental library. A possible reason is that predicted spectra can exceed the quality of some spectra in the original experimental library, e.g., those with incorrect identifications that did not accurately represent relative fragment ion intensities, as reported in previous studies on conventional proteomics[45] and phosphoproteomics[53,54]. Notably, the prediction accuracy of glycopeptides was limited because there have not been standards (as iRT kit for peptides[63]) for high-precision glycopeptide retention time calibration of yet. However, the fully predicted spectral library still led to a

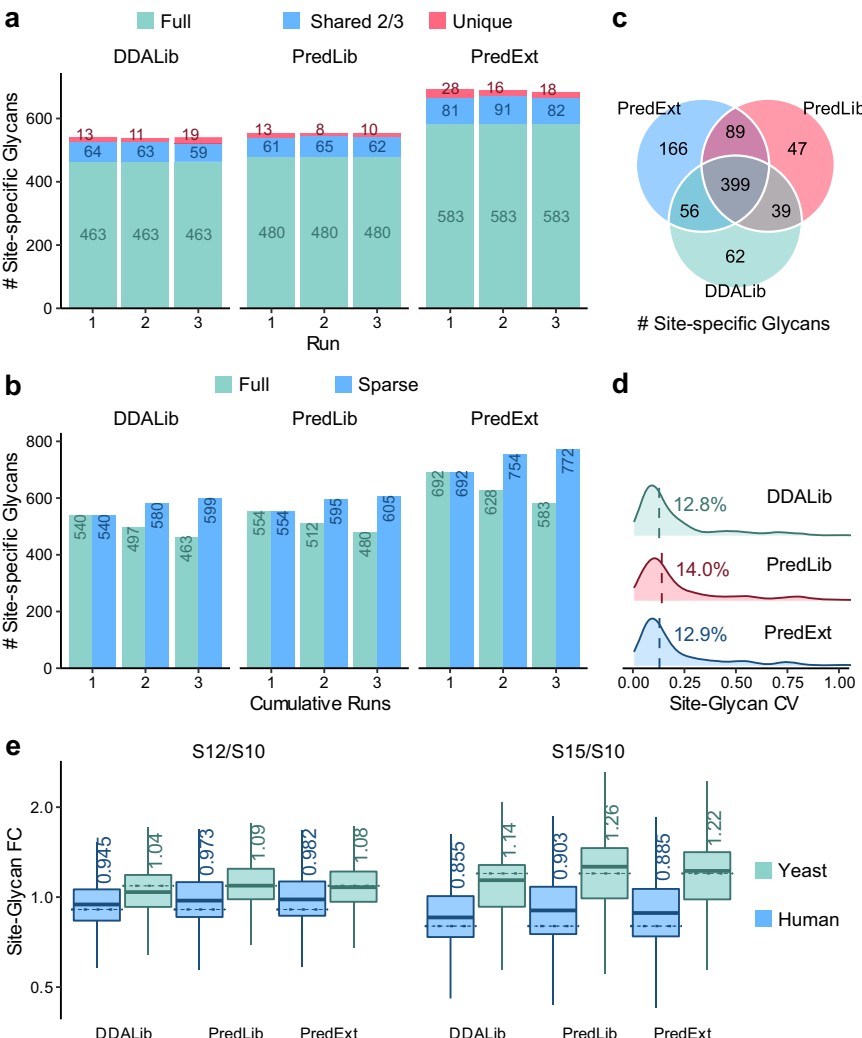

**Fig. 4 | Performance of predicted spectral libraries for DIA analysis. a** Numbers of identifications from the serum sample per run. "Full" represents identifications observed in all the runs; "shared 2/3" represents identifications observed in 2 runs; "unique" represents identifications observed in only 1 run. **b** Numbers of cumulative identifications from the serum sample across runs. "Full" represents identifications shared in the cumulative runs; "sparse" represents identifications observed in at least one run in the cumulative runs. **c** Comparison of numbers of identifications from the serum sample shared in 2/3 runs using different libraries. **d** Coefficients of variation (CVs) of quantification results. Medians are indicated.

**e** Box plot visualization of fold change of the quantification results of the mixed-organism samples. Percent changes were calculated based on the mean quantities in three replicates of each sample. The medians are indicated. The boxes indicate the interquartile ranges (IQR), and whiskers indicate 1.5 × IQR values; no outliers are shown. The dashed lines indicate theoretical fold changes of the organisms (S10:S12:S15 = 1:0.9:0.8 for human and 1:1.1:1.2 for yeast). DIA results using the predicted library (PredLib) and the extended predicted library (PredExt) are compared with results using the experimental library (DDALib) from the original publication of GproDIA. Source data are provided as a Source Data file.

close number of detected glycopeptides compared to the experimental spectral library.

The incompleteness of spectral libraries can limit the capability of detection in DIA data analysis. Therefore, we tested whether DIA analysis can benefit from predicted spectral libraries with increased coverage. Starting from glycopeptide lists of serum collected in our previous study[30], we generated an extended predicted library with ~50% more glycopeptide coverage (PredExt, containing ~5000 precursors, denoted as 5k, Supplementary Table 7) compared to the DDALib. PredExt yielded 991 ± 5 precursors of 691 ± 2 site-specific glycans per run (Fig. 4a, Supplementary Fig. 56, Supplementary Data 8). Considering identifications shared in >50% replicate runs, 24% more (1033/835) precursors and 28% more (710/556) site-specific glycans were detected using PredExt than those using DDALib. We further tested a series of predicted spectral libraries with increasing coverage up to ~10,000 precursors (Supplementary Note 9, Supplementary Figs. 56 and 57, as well as Supplementary Data 8).

The entrapment-based benchmarking, which has been used to approximately estimate false positive identifications for DIA analysis with experimental spectral libraries[30], was also performed with predicted libraries. Glycopeptides with glycans not present in the samples were added to the original predicted libraries (Supplementary Note 10). In all the analyses, we ensured that the entrapment glycans were different from those in the original library, and kept the number of entrapment glycopeptide precursors similar to the sizes of the original library (Supplementary Tables 8 and 9). The entrapment hits in the results (Supplementary Data 9 and Supplementary Fig. 58) were considered as false positives, and we used entrapment percentage (percentage of the number of entrapment hits to the target hits) to compare the false positive rates relatively, although it did not measure the true error rates exactly. The entrapment percentage using PredLib was higher than that using DDALib (1.4% compared to 1.0%) for the yeast dataset, while it was close to that using DDALib (2.3% compared to 2.5%) for the serum dataset. An extend predicted library containing

~7000 precursors (PredExt 7k) resulted in an entrapment percentage (2.3%) closed to the PredLib, while larger library coverage (PredExt 10k, ~10,000 precursors) led to substantially higher entrapment percentages. These results indicated that large libraries were adverse to statistical control of error rates in DIA glycopeptide analysis, which is currently the main limitation of predicted spectral libraries.

For the evaluation of quantitative precision, the coefficients of variation (CV) values of quantification results were calculated among the replicate runs (Fig. 4d and Supplementary Fig. 56). The median CV values were 12% at precursor level and 13% at site-specific glycan level using PredExt (5 k), close to those using DDALib. The quantitative performance of predicted spectral libraries was further evaluated on a dataset of two-glycoproteome samples of budding yeast and human serum (Supplementary Tables 5 and 10, Supplementary Data 10, and Supplementary Figs. 59–62). Fold changes of measured abundance of glycopeptides were calculated between samples with different mixing ratios (Fig. 4e). Using the predicted spectral libraries, fold changes of human glycopeptide abundance were slightly overestimated compared to those using the DDALib, while quantification accuracy of yeast glycopeptides was close to and sometimes better than DDALib. The results indicated that predicted spectral libraries were suitable for DIA data analysis with performance comparable to experimental spectral libraries.

## Discussion

In this study, we introduce a deep neural network architecture able to predict MS/MS spectra of intact glycopeptides. The main characteristic distinguishing our method from others for peptide MS/MS prediction is the ability to process non-linear glycan structures by introducing the tree LSTM networks. While separate modules played their respective roles to extract features from the peptide and glycan moieties, they shared information with each other through feature fusion regarding the glycopeptide as a whole. Multi-task learning was adopted for predicting the whole glycopeptide spectrum, as well as the peptide and glycan fragments, aiming to fit the wide range of peak intensities of different fragment types.

Our method achieved high prediction accuracy using models trained with data originated from the same organism and instrument settings. The change of organisms and instrument settings could result in losses of the prediction performance. The generalization ability of the model was still limited by the size of training data due to the difficulty in accessing large-scale glycopeptide MS/MS datasets compared to conventional proteomics datasets. We anticipate that adding additional encoders of spectral metadata, such as instrument types and collision energy[42,45], would probably facilitate the scalability of the model for spectrum prediction in other glycoproteomics datasets from independent laboratories. At this stage, however, we did not commit to build such a universal model as a compromise due to lack of high-quality reference MS/MS data of glycopeptides covering various collision energies. As an alternative, we chose to train organism- and instrument-specific models, which can be further finetuned to improve the performance for glycopeptide spectrum prediction from different origins.

Another distinct feature of our deep learning model is that the prediction is explainable by the attention weights computed in the model. It was demonstrated that the attention weights can reflect the importance of possible cleavages in the putative fragmentation pathways of a specific glycan structure. This highlighted how our model learned underlying principles in the MS/MS fragmentation of glycopeptides. This feature allowed the differentiation of glycan structural isomers by modeling the intensity variation of peaks originated from distinct fragmentation pathways. We demonstrated that predicted spectra can be utilized for spectral library searching to ranking potential glycan structures based on a given glycopeptide composition and filter out the less possible candidates. Despite a

remaining gap towards the exact identification of glycan structures by spectral library searching alone, it can discriminate glycan structural isomers partially, like the recognition of core fucosylation. Different from methods relying on confirming the presence of characteristic ions[13], spectral library searching takes the intensity pattern of the whole spectrum into consideration, which has been proven to be effective in the identification of peptides[64] and site localization of phosphorylation[54,65]. With spectrum prediction, we resolved the limitation of spectral library searching on the incomplete library coverage of glycan structure space, and showed its potential for validating or supplementing the structural identifications of glycopeptide by other methods. We further envision that spectrum prediction may improve scoring in glycopeptide database searching and de novo sequencing[57].

Our results also demonstrate that predicted spectral libraries can be used for analyzing DIA data of glycopeptides. Predicted libraries can not only correct low-quality spectra in a sample-specific experimental spectral libraries when keeping the same glycopeptide space, but they can enlarge glycoproteome coverage with improved library completeness. The current glycopeptide-centric DIA data analysis method cannot bear an extremely large query space which contains a significant fraction of false target glycopeptides not detectable in the samples[30]. This limitation was not peculiar to glycoproteomics and actually inherited from the statistical control strategy of DIA analysis for conventional proteomics[66]. Therefore, it is not practical to use a predicted glycopeptide spectral library generated from an organism-wide proteome and glycome space. Instead, a starting glycopeptide list of interest is still need to delimit the search space at present. We envision that this issue would be resolved with critical advances in DIA data analysis for glycoproteomics, like deep learning-based scoring that are compatible with proteome-scale predicted library for conventional proteomics[52].

We expect that this work will provide a valuable deep learning resource for the glycoproteomics community with other potential applications in users' informatic workflows (Supplementary Note 11). Although it is demonstrated here in the context of N-glycoproteomics, generic architecture of our deep learning model could be adapted to spectrum prediction of O-glycopeptides. We envisage that extension of the model architecture in the future will support fragmentation techniques with other fragment ion types, such as electron-transfer dissociation[56], and analytes containing multiple glycans per glycopeptide[12], in case sufficient high-quality glycopeptide MS/MS datasets are available for model training.

## Methods

### Datasets for model training and validation

HCD MS/MS spectra of intact glycopeptides were collected from 4 datasets of mouse samples, 4 of human samples, and 1 of yeast samples. They had been acquired on Orbitrap mass spectrometers with different instrument settings: (1) Mouse1[10], Mouse4[60], Human4[30], and Yeast[30] were acquired on Orbitrap Fusion with stepped CE of $30 \pm 10$; (2) Human2[15] was acquired on Orbitrap Exploris 480 with stepped CE of $30 \pm 10$; (3) Mouse2[13], Mouse3[59], Human1[61,62], and Human3[59] were acquired on Orbitrap Fusion Lumos with individual CEs of 20 and 33. Detailed information of these datasets is shown in Supplementary Table 2 and 3. Structure-specific glycopeptide identification results of each dataset, if provided with its original publication, were directly used in this study. Otherwise, we reanalyzed the datasets using StrucGP[13] (version 1.1.1) with the default settings.

For each GPSMs, peak intensities were extracted from the experimental spectrum by matched with $m/z$ of theoretical fragment ions of intact glycopeptides in HCD with stepped CE[30]. For peptide fragments, the following ion types with charge states 1+ or 2+ were considered: (1) naked peptide backbone b and y fragment ions; (2) b/y ions with one HexNAc, denoted as b-N(1)/y-N(1); (3) b/y ions with a

residue as a result of cross-ring fragmentation on the HexNAc, denoted as b$/y$ for simplicity. For glycan fragments, naked peptide (denoted as Y0) and Y ions with charge states 1+ to 3+ were considered. For glycans other than high mannose type, fragment ions from the glycan branches at the non-reducing end (referred to as B ions, Supplementary Fig. 2) with charge state 1+ were considered. For datasets with individual CEs, fragment ions in the low and high CE spectra of the same precursor were merged by averaging the peak intensities, yielding a pseudo stepped CE spectrum. Spectra with <5 peptide b/y ions or <5 glycan Y ions were excluded. For the model with branch fragments, spectra with <2 glycan B ions were further excluded. Within each dataset, fragment ions of replicate spectra of the same glycopeptide precursor were further combined to create a consensus spectrum[30]. Finally, a non-redundant dataset was obtained, containing one spectrum per glycopeptide precursor.

## Model architecture

The model contained separate modules to process the peptide and glycan moieties of a glycopeptide and predict the corresponding part of the fragment spectrum.

For the peptide moieties, the amino acid sequence was encoded as a list of 20-dimentional one-hot vector with zeros and ones. Each PTM, if any, was represented as a 6-dimentional embedding vector to represent the numbers of H, C, N, O, S, and P atoms. For amino acid without PTMs, a 6-dimentional zero vector was used as a placeholder. The amino acid vectors and PTM vectors at corresponding sequence positions were concatenated and fed to two stacked bidirectional LSTM layers (with a hidden size of 256), followed by a dropout layer (with rate of 0.25).

For the glycan moieties, the monosaccharides were one-hot encoded as 5-dimensional one-hot vectors representing Hex, HexNAc, NeuAc, NeuGc, and Fuc. A graph (strictly speaking, a tree) was built according the glycan structure, where the monosaccharides served as the nodes (with one-hot vectors as node features) and their linkages as the edges. A tree LSTM layer (with a hidden size of 256) traversed the nodes in the bottom-up direction the glycan tree, followed by a dropout layer. The cells in a tree LSTM are similar to those of the standard LSTM, except that the standard LSTM uses the hidden state of the previous timestep, whereas the tree LSTM combines the hidden states from the child nodes by summing them up (in this study, we ignore the order of branches in a glycan) to cope with variable number of children[58]. After the bottom-up traversing, the node features contained information of the monosaccharides for each node and those sprouting at the non-reducing end.

The feature of the root node, which encoded the information of the whole glycan, was transformed by a dense layer and then added to the vector at the glycosite in the features output by the peptide LSTMs. A two-dimensional vector of precursor charge state was concatenated to the peptide features at each sequence position. The updated peptide features were processed by another two bidirectional LSTM layers and a dropout layer, followed by a dense layer and the ReLU activation function to output b/y ion intensities (with 4 dimensions, i.e., 2 charge states per ion type).

The final hidden state of the last timestep output by the first two peptide LSTM layers contained the information of the whole sequence. It was transformed by a dense layer and added to the feature of the root node in the glycan tree. The features of the original monosaccharide, those from bottom-up traversing, and the charge state vector were concatenated node-wisely, and then updated by a second tree LSTM layer that traversed the glycan tree in the top-down direction, followed by a dropout layer. This allowed each node having a comprehensive view of monosaccharides downward, upward, and in the other branches. As a results, the updated node features contained information of monosaccharides, its position in the glycan structure, as well as peptide and charge state.

Considering each potential cleavage site between two monosaccharides, we split the glycan tree into nodes lost from or retained in the fragment ions with peptide (Supplementary Fig. 1). Features of the lost nodes were summed with attention weights computed by a dense layer followed by a Softmax function[67]. The retained nodes were processed similarly. The summed features of lost and retained nodes were concatenated into a 512-dimensional vector for each cleavage site. Then a tripartite graph was built according to the putative fragmentation pathways of the glycan structure, comprising three types of nodes: (1) cleavage sites; (2) structure-specific fragments originated from a series of cleavages; (3) fragments with distinct monosaccharide composition combined from isomeric structure-specific fragments that are not distinguishable by mass. Features of the cleavage nodes were aggregated by a LSTM layer (after shuffle during training since LSTM is not inherently symmetric[68]) and the attention mechanism described above. A dense layer followed by the ReLu activation function was used to output Y ion intensities (with 3 dimensions, i.e., 3 charge states). Finally, intensities of isomeric fragments with the same monosaccharide composition were summed up. Optionally, another tripartite graph could be added to the model for predicting the intensities of B ions, which was generated from combinations of link cleavages that are different from Y ions (Supplementary Fig. 2).

The peptide features output by the last two LSTM layers were summed over the peptide length dimension. The glycan node features output by the second tree LSTM were summed. The attention mechanism described above was used for weighting the features. The summed features of peptide and glycan were concatenated and then transformed into a scalar value by a dense layer with the sigmoid activation function to predict the ratio of the peptide fragment intensity to the whole MS/MS spectrum. Finally, the peptide and glycan fragments are merged by the ratio, forming the output glycopeptide spectrum.

## Model training

The dissimilarity between the predicted and experimental spectrum was measured by spectral angle loss[45] (SA), which is defined based on dot product (DP):

$$\mathrm{SA} = \frac{2}{\pi}\arccos\mathrm{DP} = \frac{2}{\pi}\arccos\frac{\boldsymbol{s}_1 \cdot \boldsymbol{s}_2}{|\boldsymbol{s}_1||\boldsymbol{s}_2|} \tag{1}$$

where $\boldsymbol{s}_1$ and $\boldsymbol{s}_2$ are the intensity vectors of the predicted and experimental spectrum, respectively. The SA and DP metrics perform an inherent L2 normalization on the intensities. SA were computed for the whole glycopeptide spectrum, as well as the peptide and glycan fragments separately. Mean square error (MSE) was used to measure the prediction error of the fragment intensity ratio. The total loss function was a weighted sum of the four objective functions:

$$L = w_1 \cdot \mathrm{SA}_{total} + w_2 \cdot \mathrm{SA}_{peptide} + w_3 \cdot \mathrm{SA}_{glycan} + w_4 \cdot \mathrm{MSE}_{ratio} \tag{2}$$

Dynamic weight average[69], a multi-task learning technique, was used to adjust the weights based on the convergence rate of loss of each subtask. In brief, the weight of loss for each subtask is determined by the loss ratio of the previous two epochs, followed by a Softmax function.

We first built a model containing only the modules to process peptide sequences and predict peptide fragment intensities. The model was trained with peptide spectra collected from a large-scale dataset of HeLa proteome[70]. When training the model for glycopeptides, parameters of corresponding modules were port form the pretrained model for peptides and the first two BiLSTM layers were frozen. The dataset was split into three distinct subsets, where 3/5 were used for training the model parameters, 1/5 for validation, and the remaining 1/5 as holdout data. The validation set was used to control

for overfitting. We used the Adam optimizer and 16 samples per batch. The learning rate started from 0 – 0.001 in 5 warmup epochs, and was then scheduled by cosine annealing with warm restarts[71] (with an initial interval of 15 epochs and multiplied by 2 after each restart). For model finetuning, an initial of learning rate 0.001 was used without warmup, and iteratively reduced to 10% when the metrics had stopped improving for 5 epochs.

The model with B ions was trained based on the model without B ions, where the layers for peptide fragment ions and glycan Y ions were frozen. The total loss function further included the SA of B ions. The learning rate was 0.0001, scheduled by warmup and cosine annealing with warm restarts in the same manner as the model without B ions.

### Differentiating structural isomeric glycopeptides by spectral library searching

For each GPSM, candidate glycopeptides were generated replacing the original glycan with its structural isomers. In this study, the built-in glycan databases of pGlyco[14] were used as the glycan space (2922 glycan structures for human and 7878 for mouse), appended with glycan structures uniquely present in the original identification results by StrucGP[13]. The fragment ions were extracted from the experimental (query) spectrum by matching to the $m/z$ of theoretical fragments of the candidate glycopeptides. Peaks absent in all of the predicted spectra for different candidate glycopeptides were discarded. SA was calculated between the fragment intensities of the query spectrum and each candidate predicted spectrum. The total spectral similarity score was combined from SA of glycan Y ions and B ions:

$$\text{Score} = \alpha \cdot (1 - \text{SA}_Y) + \beta \cdot (1 - \text{SA}_B) \qquad (3)$$

where the weights $\alpha$ and $\beta$ were set as 0.5 in this study. The candidate glycopeptides were then ranked by the similarity scores in descending order.

### DIA data analysis

Predicted spectral libraries were generated using the glycopeptide list from the DDA-based experimental spectral libraries (Supplementary Table 6). Two types of predicted spectral libraries were built: (1) only the fragment ion intensities were predicted, while experimental retention time values in the original DDA library was kept; (2) both the fragment ion intensities and retention time values were predicted. The DIA data were analyzed by GproDIA through the workflow described in its original publication[30], including transition filtering, retention time calibration, decoy generation, feature extraction, scoring and statistical control. For the serum data, the glycoform inference was enabled.

We also built predicted spectral libraries with extended coverage compared to the original DDA libraries (Supplementary Table 7). These libraries were generated from glycopeptide lists of yeast and serum from our previous study[30]. This setting is necessary to avoid combinatorial explosion of peptides and glycans. In order to reduce the computational burden, glycoform inference was turned off in a preliminary search performed first, and then enabled with a refined search space narrowed down to the preliminary search result.

The influence of predicted spectral libraries on statistical control was evaluated by adding entrapment glycopeptide precursors to the predicted libraries[30]. In all the analyses, we ensured that the entrapment glycopeptides were not present in the sample, and kept the number of entrapment glycopeptide precursors a similar number to that of the original predicted library (Supplementary Table 8 and 9). For the yeast data, entrapment entries were glycopeptides with peptide sequence from yeast and human glycans containing Fuc or NeuAc monosaccharides. For the serum data, entrapment entries in the GproDIA publication were glycopeptides with peptide sequences from human and xylosylated glycans from *Arabidopsis thaliana*. Since our

current model do not support xylose due to lack of training data, we kept the topology of glycan structures and merely replaced xylose with NeuGc. Therefore, the entrapment entries should still be absent in the serum sample. The entrapment entries in the DIA analysis results can be regarded as false positives.

### Implementation, statistics and visualization

The deep learning models were implemented in Python (version 3.9.16, Anaconda distribution version 2022.10, https://www.anaconda.com/) with PyTorch (version 1.12.1, https://pytorch.org/) and DGL (version 1.0.1, https://www.dgl.ai/). Post-analysis statistics was conducted using R (version 4.3.1, https://www.r-project.org/). The Python packages matplotlib (version 3.6.2) and networkx (version 1.6.20), as well as the R ackages ggplot2 (version 3.4.3) and VennDiagram (version 1.7.3) were used for data visualization.

### Reporting summary

Further information on research design is available in the Nature Portfolio Reporting Summary linked to this article.

## Data availability

Raw mass spectrometry data are publicly available at the ProteomeXchange Consortium with the dataset identifiers PXD005411, PXD005413, PXD005412, PXD005553, PXD005555, PXD025859, PXD035158, PXD026629, PXD026649, PXD030804, PXD031025, and PXD023980 (see Supplementary Table 2–4 for details). Data generated in this study, including processed data for model training and testing, trained models, predicted spectral libraries, and DIA analysis results, have been deposited in the ProteomeXchange via the iProX[72] partner repository with the dataset identifiers PXD045248 or IPX0007075000. Source data are provided with this paper.

## Code availability

The source code of DeepGlyco is available at Github https://github.com/yyi17/DeepGlyco and Zenodo https://zenodo.org/records/10682893[73].

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

## Acknowledgements

This work was supported by the National Natural Science Foundation of China (22304153 to Y.Y., 22234007 and 21827806 to Q.F.), Zhejiang Provincial Natural Science Foundation of China (LQ24B050003 to Y.Y.), and the Ministry of Science and Technology of China (2021YFA1301601 to Q.F.).

## Author contributions

Y.Y. conceived the project, did all the coding work, and wrote the draft of the paper. Q.F. supervised the project and revised the paper.

## Competing interests

The authors declare no competing interest.
