## [Peer Review File · Nature Communications]

Prediction of glycopeptide fragment mass spectra by deep learningReviewer #1 (Remarks to the Author):

In their manuscript "Prediction of glycopeptide fragment mass spectra by deep learning", Yang & Fang describe DeepGlyco, a novel fragment ion intensity predictor for glycopeptides. While accurate ML-based prediction of unmodified and modified peptide fragment ion intensities has been achieved a few years ago, to my knowledge, the same has not yet been demonstrated for glycopeptides. This manuscript thus presents an exciting and timely new approach for this challenge.

I found the manuscript to be well readable and concise. The corresponding source code and technical documentation provides all necessary information to make best use of DeepGlyco. I have only a few minor comments:

1. Tutorial for fine-tuning / transfer learning using independent data

DeepGlyco will certainly be one of the more complex models used for fragmentation prediction and is also trained on fewer data points, due to the nature of the scientific problem. In this regard, it is extremely important that potential users are aware of the challenges when applying the model to their own data, be it DDA rescoring or DIA quantification. Although the authors have demonstrated the potential of fine-tuning of DeepGlyco in Fig. 2, it would be great if more details or even a tutorial (e.g. as part of the GitHub repo) could be added that describe what kind of data (and amount) potential users should provide, how to fine-tune a model and which performance metrics to assess and judge, as well as what to expect when applying a predictor to a new biological problem.

2. Model overview & Running time

The authors should provide a brief overview of model complexity (e.g. number of parameters for different model components, or model card (<https://huggingface.co/blog/model-cards>)) as part of the results section. Further, estimated training and running times for selected analyses to allow potential users evaluation of the computational requirements for their target workflows would be a useful extension.

Reviewer #2 (Remarks to the Author):

This paper presents DeepGlyco, a deep learning model for predicting tandem mass spectra of intact glycopeptides. The model uses LSTM and tree-structured LSTM networks to process the peptide sequence and glycan structure respectively. Evaluation results show the method achieves high accuracy in predicting glycopeptide spectra across different sample types and instruments. It can also partially distinguish between glycan structural isomers based on differences between their predicted spectra.

Overall assessment:

1. Overall, this is a well-written manuscript presenting an innovative deep learning approach to glycopeptide identification using tandem mass spectra. The model design is thoughtful and technically solid.

2. The evaluations are fairly extensive over multiple datasets. The results demonstrate promising performance. One limitation is that the generalization of the model across different datasets is still limited, requiring retraining or fine-tuning. More rigorous cross-dataset testing would be informative, and helpful for understanding the source of the variations of glycopeptide fragmentation.

Major Suggestions/Comments:

1. The authors generated consensus spectra for the testing purpose, but it may not be the most appropriate approach. It would be more informative to evaluate similarities by reporting predicted similarity against each replicate, which will provide insight into the model's stability over

replicates, and may potentially reveal a larger standard deviation in the distribution of the resulting similarity.

2. The authors employed a sub-node summing variant of tree-LSTM that doesn't consider the order of sub-nodes. It would be valuable if the authors could explain their rationale for choosing this approach over other variations of tree-LSTM that can preserve sub-node orders.

3. The paper mentions the use of the pretrained peptide part. It would be helpful to quantify the improvement achieved through the pertaining.

4. The paper reports similarity results using the models trained on individual datasets. While the authors discuss the option of training organism- and instrument-specific models, it is worth exploring whether combining all datasets to create a larger dataset would lead to better generalization.

5. In the final stage of the model, where the peptide and glycan fragments are merged by ratio, it's crucial to investigate whether this predicted ratio may be biased towards glycan ions as glycan ions are much more intensive. Also, a more in-depth exploration of how this ratio affects the overall similarity is warranted.

Minor Comments:

1. For similarity calculation, it would be beneficial to clarify whether intensities are normalized in any manner before computing similarities.

2. Additional details on how the Dynamic Weight Average is implemented would improve the paper's clarity.

3. The paper mentioned in the discussion section "it incorporated potential fragmentation mechanisms of glycopeptides in MS/MS, which is explainable rather than a black box". It's advisable to avoid such statements as the paper doesn't provide a section on how attention weights can be used for mechanistic explanations.

4. To differentiate structural isomers, it would be insightful to analyze if there are trends indicating which types of glycans are more challenging to distinguish via spectral matching. A more detailed analysis of the types of misidentifications could offer valuable insights on this issue.

5. In the method section, the paper adds another tripartite graph for predicting the intensities of B ions. Please explain why the intensities of B ions cannot be predicted using the same tripartite graph?

Reviewer #3 (Remarks to the Author):

Reviewer #4 (Remarks to the Author):

In this manuscript, the authors have developed a deep learning model for predicting the fragmentation spectra of intact N-glycopeptides and applied it to refine glycan structure identifications and generate a spectral library for glycopeptide DIA analysis. Methods for high quality prediction of spectra and retention times of glycopeptides would enable dramatic improvements to both DDA and DIA analyses of glycopeptides, making such predictions a very

important but extremely challenging problem in glycoproteomics. The authors have made an impressive effort to tackle many of the challenges inherent in these predictions and have produced a model that appears to generate high-quality predicted spectra when trained on data from the same experiment. However, both use cases to which the authors apply the predicted spectra, identifying glycan structures and analyzing DIA data, show limited success using the predictions. The interpretation and discussion of these results should also be revised to explicitly state or expand on several key limitations and caveats.

Specific Comments:

Major:

- The glycopeptides from 5 mouse tissues dataset used in training the prediction model (from Liu et al, 2017) are extensively adducted with ammonium, but this was not included in the StrucGP search used to identify the glycopeptides prior to training. As a result, a proportion of the glycopeptides likely have incorrectly assigned compositions with excessive fucoses (for example, hexose + fucose has a mass difference of 0.01 Da with NeuAc + ammonium, and a high mannose glycan with an ammonium adduct can also have a nearly identical mass to a hypothetical complex glycan with 3 or 4 fucoses). In Supplementary Data 1, for example, over 10% of the GPSMs are identified with unlikely compositions with 3 or 4 fucoses (8% when considering rank 1 only). This is problematic for several reasons, but particularly in Figure 3b (when assessing the accuracy of the spectral library search for glycan structural isomers), a match to the same glycan after rescoring is shown as a "correct" identification in the figures when in fact some of them are incorrect at both the composition and structure levels. Ideally, these GPSMs would be corrected to provide accurate training data, or at least excluded from the training.
- Even after excluding or correcting common cases of incorrect glycan assignments, there will remain some incorrectly identified glycan compositions, and especially structures, in the training data. Can the authors comment on whether this is accounted for at all in the deep learning method, and what impact it will have on the resulting predictions?
- The FUT8 knockout and exoglycosidase treatment samples in Fig 3 provide good insight into the structure assignment error rates. However, the manuscript text should explicitly state that the majority of the 12% of the original and 5% of rescored core fucosylated GPSMs from the FUT8 KO sample are likely false identifications. This still indicates improvement with the rescoring (a 7% reduction in false core fucosylation), but the limitations must clearly be stated since the estimated FDR is supposed to be 1%.
- The authors mention that the prediction method requires instrument-specific training as no instrument or collision energy metadata is included in the model. I believe this means that for anyone other than the authors to use the method, they must retrain the model from scratch using their own glycopeptide LC-MS data. This results in a major limitation for the community as it appears that only StrucGP (for DDA) and GproDIA (for DIA) software tools are supported for the initial analysis of training or library-building data, respectively. These tools are not the most common software tools used for glycoproteomics data analysis - could other search engines' results be converted to use with DeepGlyco (and would be the key information needed to do so)? If not, how can other glycoproteomics researchers make use of the method?
- It is not clear to me that the predicted MS2 spectra and retention times add value to the DIA analysis. The performance was slightly worse (yeast dataset) or roughly the same (serum dataset) compared to the original analysis of GproDIA without predictions. Since the DDA library is still required to perform the initial search with GproDIA, the typical advantage for predicted libraries in conventional proteomics (not having to perform a DDA experiment) is not applicable here. What is the use case for the predictions vs simply using GproDIA?
- The extended library provided improved performance for the DIA analysis, but it was not compared to simply providing an extended library to GproDIA without the spectrum predictions. My understanding is that the extended library was again generated from previous DDA data (rather than predicted fully in-silico), so it seems that it could have been used in GproDIA without the prediction method to achieve similarly improved performance. If this is the case, what is the

value being added by the prediction method? Or if not, please clarify in the methods exactly how the extended library was generated.

- The entrapment searches used to provide an empirical FDR for glycan composition assignments in the DIA analyses are too simple to provide a reasonable estimate of the composition-level FDR. In the fission yeast dataset, excluding glycan compositions with fucose and sialic acid is (or should be) trivial as there are no glycans with those monosaccharides in the yeast and thus no spectra should contain any specific fragment ions. The key challenge that needs to be assessed for glycopeptide DIA FDR is the ability of the method to correctly assign the glycan composition of a glycopeptide in the presence of glycan fragments from other glycopeptides in the chimeric DIA spectrum. If there are no NeuAc-specific oxonium ions in any glycan on any glycopeptide, for example, then not assigning any glycans with NeuAc is trivial and the method should appear to provide a perfect empirical FDR in the entrapment search. But in the human serum dataset, or any other dataset with NeuAc-containing glycans, correctly assigning the glycan compositions is not trivial as many spectra will have NeuAc-oxonium ions from some, but not necessarily all, glycopeptides in the spectrum. The accuracy of glycan compositions assigned in these cases is never assessed in this framework. The entrapment searches should be revised to include glycans with shared monosaccharides but nonsensical or not-present compositions to provide an empirical FDR assessment, and these limitations should be discussed in the text.

o This is particularly important for assessing the FDR of the extended library. My assumption is that adding glycopeptides with glycans that contain NeuAc and/or Fuc but are not known in humans to an extended library would result in a substantially higher entrapment rate (though I would be happy to be proven wrong).

Minor:

- The methods section states that the order of glycan branches is ignored, which is understandable given the limitations of glycan structure assignment from a single glycopeptide HCD spectrum. However, the branching order can have biological implications, so this caveat needs to be provided when describing the results as having determined the “correct structure” or the method as being “structure specific”.

- The text describing the DIA results claims that “in line with previous studies on conventional peptides and phosphopeptides...the library with predicted spectrum performed better”, however, the serum library with RT prediction was not better and neither of the yeast libraries with prediction performed better. In conventional proteomics and phosphoproteomics, retention time prediction typically provides a large performance boost, unlike the effect seen here. This statement should be revised to more accurately reflect the data presented.

- Several of the Supplementary Data tables do not contain the protein ID for the identified GPSMs, making it hard to assess the results, particularly for the analysis of the glycoprotein standards. Please consider including this information to the tables.

Responses to the Reviewer's Comments

Prediction of glycopeptide fragment mass spectra by deep learning

Yi Yang et al.

Revision Summary

During the revision, we have:

- (1) Explored the model generalization trained with a larger combined dataset;
- (2) Evaluated the prediction similarities on replicate spectra and investigated the impact of predicted intensity ratio on the overall similarity, providing further evidence on the origin of the variation in glycopeptide fragmentation;
- (3) Investigated the impact of potential incorrect identification in the training data, re-trained and re-evaluated the model with a clean dataset;
- (4) Performed additional DIA analysis using extended spectral libraries with larger coverage and the stricter entrapment strategy proposed by the reviewer. The results clearly show the advantage of the method presented beyond earlier works.

All the corresponding changes have been highlighted in the marked revised manuscript. Our point-by-point responses to each reviewer's comments are present below.

Reviewer #1:

In their manuscript "Prediction of glycopeptide fragment mass spectra by deep learning", Yang & Fang describe DeepGlyco, a novel fragment ion intensity predictor for glycopeptides. While accurate ML-based prediction of unmodified and modified peptide fragment ion intensities has been achieved a few years ago, to my knowledge, the same has not yet been demonstrated for glycopeptides. This manuscript thus presents an exciting and timely new approach for this challenge.

I found the manuscript to be well readable and concise. The corresponding source code and technical documentation provides all necessary information to make best use of DeepGlyco. I have only a few minor comments:

Response: Thank you very much for taking the time and effort to review our manuscript. We really appreciate the valuable comments and suggestions, and have modified our manuscript accordingly.

1. Tutorial for fine-tuning / transfer learning using independent data

DeepGlyco will certainly be one of the more complex models used for fragmentation prediction and is also trained on fewer data points, due to the nature of the scientific problem. In this regard, it is extremely important that potential users are aware of the challenges when applying the model to their own data, be it DDA rescoring or DIA quantification. Although the authors have demonstrated the potential of fine-tuning of DeepGlyco in Fig. 2, it would be great if more details or even a tutorial (e.g. as part of the GitHub repo) could be added that describe what kind of data (and amount) potential users should provide, how to fine-tune a model and which performance metrics to assess and judge, as well as what to expect when applying a predictor to a new biological problem.

Response: Thank you for the suggestions. In the revision, we provided a tutorial of model finetuning as part of the code repo.

2. Model overview & Running time

The authors should provide a brief overview of model complexity (e.g. number of parameters for different model components, or model card (<https://huggingface.co/blog/model-cards>)) as part of the results section. Further, estimated training and running times for selected analyses to allow potential users evaluation of the computational requirements for their target workflows would be a useful extension.

Response: Thank you for the suggestions. In the revision, we provided a brief overview of model complexity in the Results section (on page 6). Estimated running times were present in **Supplementary Note 11**:

“ The model without B ions contains 7,111,440 parameters in total (5,456,396 for the peptide part, 1,653,507 for the glycan part, and 1537 for the intensity ratio). The model with B ions contains 7,900,433 parameters in total (additional 788,993 for the glycan B ions). In this study, model training was performed on a workstation with an Intel Core i9-12900K CPU, 64 GB RAM, and a NVIDIA GeForce RTX 3090 GPU (24 GB memory). Training the model (470 epochs) without B ions using the Mouse 1 (12,818 spectra) and Human 1 (19,137 spectra) dataset took ~20 h and ~41h, respectively. Finetuning the model (20 epochs) with Mouse 2 (5942 spectra) and Human 2 (3364 spectra) took ~21 min and ~14 min, respectively. Training the model (470 epochs) with B ions with Mouse 1 (6202 spectra) and Human 1 (15,042 spectra) took ~14 h and ~33 h, respectively. Predicting ~10,000 glycopeptide spectra took ~5 min. Users can evaluate the computational requirements for integrating DeepGlyco in their informatics workflows.

Reviewer #2:

This paper presents DeepGlyco, a deep learning model for predicting tandem mass spectra of intact glycopeptides. The model uses LSTM and tree-structured LSTM networks to process the peptide sequence and glycan structure respectively. Evaluation results show the method achieves high accuracy in predicting glycopeptide spectra across different sample types and instruments. It can also partially distinguish between glycan structural isomers based on differences between their predicted spectra.

Overall assessment:

1. Overall, this is a well-written manuscript presenting an innovative deep learning approach to glycopeptide identification using tandem mass spectra. The model design is thoughtful and technically solid.
2. The evaluations are fairly extensive over multiple datasets. The results demonstrate promising performance. One limitation is that the generalization of the model across different datasets is still limited, requiring retraining or fine-tuning. More rigorous cross-dataset testing would be informative, and helpful for understanding the source of the variations of glycopeptide fragmentation.

Response: Thank you very much for taking the time and effort to review our manuscript. We really appreciate the valuable comments and suggestions, and have modified our manuscript accordingly.

Major Suggestions/Comments:

1. The authors generated consensus spectra for the testing purpose, but it may not be the most appropriate approach. It would be more informative to evaluate similarities by reporting predicted similarity against each replicate, which will provide insight into the model's stability over replicates, and may potentially reveal a larger standard deviation in the distribution of the resulting similarity.

Response: Thank you for the suggestions. We used consensus spectra for model training because the number of replicates varies among different glycopeptides and a small portion of replicate spectra may be incorrect identification. In the revision, we reported prediction similarities against both the consensus spectra and each replicate spectra. Results evaluated on the replicate spectra are present in supplementary figures next to those of consensus spectra. To explain the larger deviation in the similarity distribution over replicates, we further provided example spectra (**Supplementary Data 1**) showing the variation in glycopeptide fragmentation among replicates.

An example figure is present below:

Supplementary Fig. 3. Performance of glycopeptide fragment spectrum prediction evaluated on all replicate spectra.

(a) Distributions of spectral similarities between predicted and experimental fragment ion intensities for glycopeptides contained in the training or holdout set of Mouse 1 and Human 1. (b) Distributions of spectral similarities for Mouse 2 and Human 2 after model finetuning. Related to **Fig. 2a** and **2c**. The center lines indicate the median values. The lower/upper hinges of the boxes indicate the first/third quartiles, and the lower/upper whiskers extend from the hinges to the smallest/largest value no further than 1.5 times the interquartile range. The data are grouped by the number of replicate spectra of each glycopeptide precursor, and the number of spectra (n) in each group is indicated. Source data are provided as a Source Data file.

2. The authors employed a sub-node summing variant of tree-LSTM that doesn't consider the order of sub-nodes. It would be valuable if the authors could explain their rationale for choosing this approach over other variations of tree-LSTM that can preserve sub-node orders.

Response: Thank you for the suggestions. We ignored the order of glycan branches because this information is hardly provided by the conventional HCD MS/MS-based glycopeptide identification method, and thus absent in the datasets used in this study.

In the revision, we explicitly stated this rationale (on page 5).

3. The paper mentions the use of the pretrained peptide part. It would be helpful to quantify the improvement achieved through the pretraining.

Response: Thank you for the suggestions. In the original submission, we used the pretrained peptide part based on an intuition to make use of the available peptide datasets, which are much more sufficient than glycopeptide datasets. It is also a common practice in deep learning to use pretrained layers as a feature extractor. In the revision, we evaluated the performance of models trained without the use of a pretrained peptide part, where the prediction similarities were close to those using the pretrained peptide part (**Response Fig. 1 and 2**). Thus, we have avoided any claim suggesting “performance improvement” through pretraining. The use of the pretrained peptide part was recorded in the Methods section for reproducing the results.

Response Fig. 1. Performance of glycopeptide fragment spectrum prediction using models without pretrained peptide part evaluated on consensus spectra.

(a) Distributions of spectral similarities between predicted and experimental fragment ion intensities, where spectra are predicted using a model trained with the Mouse 1 dataset. (b) Result using the model trained with the Human 1 dataset. (c) Distributions of prediction error of the intensity ratio using the Mouse 1 model. (d) Prediction error of the intensity ratio using the Human 1 model. The median values of spectral angle loss (SA) and dot product (DP), as well as data size (n) and mean squared error (MSE), are indicated. The lower/upper hinges of the boxes indicate the first/third quartiles, and the lower/upper whiskers extend from the hinges to the smallest/largest value no further than 1.5 times the interquartile range.

Response Fig. 2. Performance of glycopeptide fragment spectrum prediction using models without pretrained peptide part evaluated on all replicate spectra.

(a) Distributions of spectral similarities between predicted and experimental fragment ion intensities, where spectra are predicted using a model trained with the Mouse 1 dataset. (b) Result using the model trained with the Human 1 dataset. (c) Distributions of prediction error of the intensity ratio using the Mouse 1 model. (d) Prediction error of the intensity ratio using the Human 1 model. Related to **Response Fig. 1**. The center lines indicate the median values. The lower/upper hinges of the boxes indicate the first/third quartiles, and the lower/upper whiskers extend from the hinges to the smallest/largest value no further than 1.5 times the interquartile range. The data are grouped by the number of replicate spectra of each glycopeptide precursor, and the number of spectra (n) in each group is indicated.

4. The paper reports similarity results using the models trained on individual datasets. While the authors discuss the option of training organism- and instrument-specific models, it is worth exploring whether combining all datasets to create a larger dataset would lead to better generalization.

Response: Thank you for the suggestions. In the revision, we trained a model using a larger dataset combined from six datasets of mouse and human (i.e., Mouse 1, Mouse 3, Mouse 4, Human 1, Human 3, and Human 4). The combined dataset (consensus spectra) was randomly partitioned into 4/5 involved in training and 1/5 holdout. The model achieved high prediction accuracy for the six datasets (with the median SA values of 0.25–0.17 for the peptide part, 0.18–0.13 for the glycan part, and 0.22–0.15 for the whole spectrum, corresponding to DP of 0.93–0.96, 0.96–0.98, and 0.94–0.97, respectively, for the holdout subsets). The model was then evaluated on the remaining three datasets (i.e., Mouse 2, Human 2, and Yeast). On the Mouse 2 and Human 2 datasets, the model outperformed the models trained using the Mouse 1 or Human 1 dataset alone. The median SA was 0.27–0.22 for the peptide part, 0.21–0.19 for the glycan part, and 0.26–0.21 for the whole spectrum, corresponding to DP of 0.91–0.94, 0.94–0.96, and 0.92–0.95, respectively. On the Yeast dataset, the prediction similarities (with the median SA of 0.29 for the peptide part, 0.13 for the glycan part, and 0.19 for the whole spectrum, corresponding to DP of 0.90, 0.98, and 0.95, respectively) were close to those using the model trained by Mouse 1. These results suggest that incorporating more datasets of different instruments or organisms for model training would lead to better generalization in the future.

The results are presented in **Supplementary Note 3** and **Supplementary Figs. 18–21, 28** and **29**. Since only limited datasets are available currently, we still trained/finetuned and evaluate the models on individual datasets for benchmarking the model stability across instruments and organisms.

Supplementary Fig. 18. Performance of glycopeptide fragment spectrum prediction using a model trained with a combined dataset evaluated on consensus spectra.

(a) Distributions of spectral similarities between predicted and experimental fragment ion intensities for glycopeptides contained in the six datasets that have been merged into the combined dataset. (b) Results tested on the other three datasets. Instrument settings of each dataset are marked. Spectral similarities are computed for peptide b/y ions and glycan Y ions separately, as well as for the total spectrum of peptide and glycan ions. The median values of spectral angle loss (SA) and dot product (DP), as well as data size (n), are indicated. The lower/upper hinges of the boxes indicate the first/third quartiles, and the lower/upper whiskers extend from the hinges to the smallest/largest value no further than 1.5 times the interquartile range. Source data are provided as a Source Data file.

5. In the final stage of the model, where the peptide and glycan fragments are merged by ratio, it's crucial to investigate whether this predicted ratio may be biased towards glycan ions as glycan ions are much more intensive. Also, a more in-depth exploration of how this ratio affects the overall similarity is warranted.

Response: Thank you for the suggestions. In the revision, we visualized of prediction error of the intensity ratio between peptide and glycan fragments (**Supplementary Figs. 18–21, 28 and 29**). Evaluated on the same datasets used for model training, the predicted ratio was not biased on the holdout subset. For cross-dataset evaluation, the model trained with Mouse 1 underestimated the ratio on some datasets and overestimated it on others. The model trained with Human 1 underestimated the ratio (biased towards glycan ions) on most of the datasets, while models trained with Mouse 2 or Human 2 overestimated the ratio (biased towards peptide ions) on some datasets. The results also suggest the variation in glycopeptide fragmentation among these datasets.

In order to explore how this ratio affects the overall spectral similarity, we intentionally introduced an “error” by adding an offset to the predicted ratio value before merging the peptide and glycan fragments into the whole predicted spectra. The spectral similarities decreased with the absolute value of ratio offset growing (**Supplementary Figs. 10 and 11**). The spectral similarities were less susceptible to negative ratio error as glycan fragments are more intensive.

The results are presented in **Supplementary Note 1**.

Supplementary Fig. 8. Prediction error of intensity ratio between peptide and glycan fragments evaluated on consensus spectra.

(a) Distributions of prediction error between predicted and experimental intensity ratio values using the model trained with Mouse 1. (b) Results predicted using the model trained with Human 1. The mean squared error (MSE) and data size (n) are indicated. The lower/upper hinges of the boxes indicate the first/third quartiles, and the lower/upper whiskers extend from the hinges to the smallest/largest value no further than 1.5 times the interquartile range. Source data are provided as a Source Data file.

Supplementary Fig. 16. Prediction error of intensity ratio between peptide and glycan fragments using finetuned models evaluated on consensus spectra.

(a) Distributions of prediction error between predicted and experimental intensity ratio values, where spectra are predicted using the model trained with Mouse 1 and finetuned with Mouse 2. (b) Results predicted using the model trained with Human 1 and finetuned with Human 2. The mean squared error (MSE) and data size (n) are indicated. The lower/upper hinges of the boxes indicate the first/third quartiles, and the lower/upper whiskers extend from the hinges to the smallest/largest value no further than 1.5 times the interquartile range. Source data are provided as a Source Data file.

Supplementary Fig. 10. Impact of intensity ratio on performance of glycopeptide fragment spectrum prediction evaluated on consensus spectra.

(a) Distributions of spectral similarities between predicted and experimental fragment ion intensities when an offset is added to the predicted intensity ratio, tested on the Mouse 1 dataset. (b) Results tested on the Human 1 dataset. The median values of spectral angle loss (SA) and dot product (DP), as well as data size (n), are indicated. The lower/upper hinges of the boxes indicate the first/third quartiles, and the lower/upper whiskers extend from the hinges to the smallest/largest value no further than 1.5 times the interquartile range. Source data are provided as a Source Data file.

Minor Comments:

1. For similarity calculation, it would be beneficial to clarify whether intensities are normalized in any manner before computing similarities.

Response: Thank you for the suggestions. The spectral angle loss (SA) and dot product (DP) metrics perform an inherent L2 normalization on the intensities.

$$SA = \frac{2}{\pi} \arccos DP = \frac{2}{\pi} \arccos \frac{\mathbf{s}_1 \cdot \mathbf{s}_2}{|\mathbf{s}_1||\mathbf{s}_2|} \quad (1)$$

In the revision, we explicitly clarified this point after the equation in the Methods section (on page 21).

2. Additional details on how the Dynamic Weight Average is implemented would improve the paper's clarity.

Response: Thank you for the suggestions. Dynamic weight average adjusts the weights based on the convergence rate of loss of each subtask. In brief, the weight of loss for each subtask is determined by the loss ratio of the previous two epochs, followed by a SoftMax function. In the revision, we added a brief description of the dynamic weight average strategy in the Methods section (on page 22).

3. The paper mentioned in the discussion section "it incorporated potential fragmentation mechanisms of glycopeptides in MS/MS, which is explainable rather than a black box". It's advisable to avoid such statements as the paper doesn't provide a section on how attention weights can be used for mechanistic explanations.

Response: Thank you for the suggestions. In the revision, we removed these statements.

4. To differentiate structural isomers, it would be insightful to analyze if there are trends indicating which types of glycans are more challenging to distinguish via spectral matching. A more detailed analysis of the types of misidentifications could offer valuable insights on this issue.

Response: Thank you for the suggestions. The conflicting results of spectral library searching with the original StrucGP annotations were counted according to the misidentified branch types (**Supplementary Figs. 31, 33 and 35**). In the revision, we described representative types of conflicting identifications in **Supplementary Note 4**.

5. In the method section, the paper adds another tripartite graph for predicting the intensities of B ions. Please explain why the intensities of B ions cannot be predicted using the same tripartite graph?

Response: Thank you for the comment. As shown in **Supplementary Figs. 1 and 2**, the Y and B ions are generated from different combinations of link cleavages, and thus they are predicted using different tripartite graphs. In the revision, we explicitly stated this reason in the Methods section (on page 21).

Reviewer #3:

Reviewer #4:

In this manuscript, the authors have developed a deep learning model for predicting the fragmentation spectra of intact N-glycopeptides and applied it to refine glycan structure identifications and generate a spectral library for glycopeptide DIA analysis. Methods for high quality prediction of spectra and retention times of glycopeptides would enable dramatic improvements to both DDA and DIA analyses of glycopeptides, making such predictions a very important but extremely challenging problem in glycoproteomics. The authors have made an impressive effort to tackle many of the challenges inherent in these predictions and have produced a model that appears to generate high-quality predicted spectra when trained on data from the same experiment. However, both use cases to which the authors apply the predicted spectra, identifying glycan structures and analyzing DIA data, show limited success using the predictions. The interpretation and discussion of these results should also be revised to explicitly state or expand on several key limitations and caveats.

Response to general comments: Thank you very much for taking the time and effort to review our manuscript. We really appreciate the valuable comments and suggestions, and have modified our manuscript accordingly.

Specific Comments:

Major:

- The glycopeptides from 5 mouse tissues dataset used in training the prediction model (from Liu et al, 2017) are extensively adducted with ammonium, but this was not included in the StrucGP search used to identify the glycopeptides prior to training. As a result, a proportion of the glycopeptides likely have incorrectly assigned compositions with excessive fucoses (for example, hexose + fucose has a mass difference of 0.01 Da with NeuAc + ammonium, and a high mannose glycan with an ammonium adduct can also have a nearly identical mass to a hypothetical complex glycan with 3 or 4 fucoses). In Supplementary Data 1, for example, over 10% of the GPSMs are identified with unlikely compositions with 3 or 4 fucoses (8% when considering rank 1 only). This is problematic for several reasons, but particularly in Figure 3b (when assessing the accuracy of the spectral library search for glycan structural isomers), a match to the same glycan after rescoring is shown as a “correct” identification in the figures when in fact some of them are incorrect at both the composition and structure levels. Ideally, these GPSMs would be corrected to provide accurate training data, or at least excluded from the training.

Response: Thank you for the suggestions. In the revision, we removed glycopeptide spectra with more than two fucoses in the Mouse 1 dataset. The cleaned dataset was then used for model training and evaluation, as well as spectral library searching. The data and figures were updated accordingly.

- Even after excluding or correcting common cases of incorrect glycan assignments, there will remain some incorrectly identified glycan compositions, and especially structures, in the training data. Can the authors comment on whether this is accounted for at all in the deep learning method, and what impact it will have on the resulting predictions?

Response: Thank you for the comment. In the revision, we created a “dirty” dataset by intentionally disturbing 5% of GPSMs in the Mouse 1 datasets. These GPSMs were randomly selected from those containing Hex and Fuc, or containing NeuAc. For each GPSM, the glycan annotation was changed to another glycan with one more NeuAc, one fewer Hex and one fewer Fuc than the original glycan, or inversely. Following the same data preprocessing procedure (including consensus spectra generation), the dirty dataset was used to training a model, which was then evaluated on the original Mouse 1 dataset (**Supplementary Figs. 12 and 13**). Compared with the original model, the dirty model performed slightly worse for the prediction of glycan fragment intensities and the whole spectrum. The prediction accuracy for the glycopeptides related to the disturbed GPSMs was affected. These results suggest the importance of quality control of training data. We hope that the community could provide more high-quality glycoproteomic datasets in the future, which would lead to better performance for spectrum prediction.

The results are presented in **Supplementary Note 2**.

Supplementary Fig. 12. Impact of incorrectly identified spectra in the training data on performance of glycopeptide fragment spectrum prediction evaluated on consensus spectra.

(a) Distributions of spectral similarities between predicted and experimental fragment ion intensities, where spectra are predicted using a model trained with a dirty dataset (containing intentional incorrect GPSMs) and compared to experimental spectra in the Mouse 1 dataset. (b) Performance comparison between the models trained with the original Mouse 1 dataset and the dirty dataset, evaluated on the glycopeptides related to the incorrect GPSMs. (c) Distributions of prediction error of the intensity ratio using the dirty model, evaluated on the Mouse1 dataset. (d) Prediction error of the intensity ratio evaluated on the glycopeptides related to the incorrect GPSMs. The median values of spectral angle loss (SA) and dot product (DP), as well as data size (n) and mean squared error (MSE), are indicated. The lower/upper hinges of the boxes indicate the first/third quartiles, and the lower/upper whiskers extend from the hinges to the smallest/largest value no further than 1.5 times the interquartile range. Source data are provided as a Source Data file.

- The FUT8 knockout and exoglycosidase treatment samples in Fig 3 provide good insight into the structure assignment error rates. However, the manuscript text should explicitly state that the majority of the 12% of the original and 5% of rescored core fucosylated GPSMs from the FUT8 KO sample are likely false identifications. This still indicates improvement with the rescoring (a 7% reduction in false core fucosylation), but the limitations must clearly be stated since the estimated FDR is supposed to be 1%.

Response: Thank you for the comment. In the revision, we clarified that the core fucosylated GPSMs were likely false identifications. We also explicitly stated the limitations on glycan structure assignment error rates (on page 12):

“ The majority of the remaining core fucosylated GPSMs were likely false identifications. The challenge of accurate glycan structure identification has not been fully resolved since the estimated false discovery rate was supposed to be 1%. Nevertheless, the results still indicate improvement with the spectrum prediction-based rescoring.

- The authors mention that the prediction method requires instrument-specific training as no instrument or collision energy metadata is included in the model. I believe this means that for anyone other than the authors to use the method, they must retrain the model from scratch using their own glycopeptide LC-MS data. This results in a major limitation for the community as it appears that only StrucGP (for DDA) and GproDIA (for DIA) software tools are supported for the initial analysis of training or library-building data, respectively. These tools are not the most common software tools used for glycoproteomics data analysis - could other search engines' results be converted to use with DeepGlyco (and would be the key information needed to do so)? If not, how can other glycoproteomics researchers make use of the method?

Response: Thank you for the comment. Although StrucGP (for DDA) and GproDIA (for DIA) was used in this study, the application of DeepGlyco is not limited to them. To prepare the data of model training/finetuning, glycopeptide MS/MS spectra with confidently identified glycan structures are needed. In addition to StrucGP, pGlyco also reports plausible glycan structures in canonical form (without strict structure-specific quality control), while most of the other common software tools for glycoproteomics data analysis identify the glycans at the level of monosaccharide compositions. Users can convert results from other search engines to DeepGlyco-supported format with the glycan structure information provided manually.

The discussion is presented in **Supplementary Note 11**.

- It is not clear to me that the predicted MS2 spectra and retention times add value to the DIA analysis. The performance was slightly worse (yeast dataset) or roughly the same (serum dataset) compared to the original analysis of GproDIA without predictions. Since the DDA library is still required to perform the initial search with GproDIA, the typical advantage for predicted libraries in conventional proteomics (not having to perform a DDA experiment) is not applicable here. What is the use case for the predictions vs simply using GproDIA?

Response: Thank you for the comment. A predicted spectral library should contain two levels of information: (1) which glycopeptides should be measured in a sample; (2) their fragment ions and retention time values. Both of them will affect the DIA analysis results. Therefore, we first evaluated the quality of predicted values by keeping the coverage of predicted library equivalent to the DDA library, before enlarging the library coverage. This arrangement was in line with previous studies of conventional proteomics (*Nature Methods* 2019, 16, 509-518; *Nature Communications* 2020, 11, 146; *Nature Communications* 2021, 12, 6685). The results indicate that the quality of predicted fragment ions was comparable to experimental spectral libraries. In the revision, we explained this arrangement to make the thread more clearly (on page 12 and 13).

- The extended library provided improved performance for the DIA analysis, but it was not compared to simply providing an extended library to GproDIA without the spectrum predictions. My understanding is that the extended library was again generated from previous DDA data (rather than predicted fully in-silico), so it seems that it could have been used in GproDIA without the prediction method to achieve similarly improved performance. If this is the case, what is the value being added by the prediction method? Or if not, please clarify in the methods exactly how the extended library was generated.

Response: Thank you for the comment. DeepGlyco enables spectral library prediction direct from a glycopeptide list and thus can break through the limitation of spectral library coverage by DDA experiments. This feature is different from the semi-empirical library extension method implemented in GproDIA. The latter uses a k -nearest neighbor strategy so that a target glycopeptide can be predicted only when an experimental library contains at least k glycopeptides with the same peptide sequence and at least k ones with the same glycan as the target. In contrast, DeepGlyco does not have this limitation.

To demonstrate this difference, we generated a series of predicted spectral libraries with increasing coverage, i.e., PredExt 5k (containing ~5000 glycopeptide precursors), PredExt 7k (~7000 precursors), and PredExt 10k (~10,000 glycopeptide precursors). The coverage of PredExt 5k was close to the semi-empirical library by GproDIA (DDAExt 5k). The number of detected glycopeptides by the predicted libraries grew with the increasing library coverage (**Supplementary Fig. 54 and 55**). The CV values of quantification results among the replicate runs were very close to those using DDAExt 5k.

We further evaluated the level of false positive identifications using the entrapment

strategy (**Supplementary Note 10** and **Supplementary Fig. 56**). The results demonstrate that larger library coverage was adverse to error rate control, which is currently the main limitation of predicted spectral libraries.

The results are presented in **Supplementary Note 9**. Corresponding revision was made on page 14 and 15.

Supplementary Fig. 55. DIA results of the human serum sample at the site-specific glycan level using the semi-empirical spectral library by GproDIA (DDAExt) and the extended predicted spectral libraries with increasing coverage (PredExt 5k, 7k, and 10k).

(a) Numbers of identifications per run. “Full” represents identifications observed in all the runs; “shared 2/3” represents identifications observed in 2 runs; “unique” represents identifications observed in only 1 run. (b) Numbers of cumulative identifications from run 1 to 3. “Full” represents identifications shared in the cumulative runs; “sparse” represents identifications observed in at least one run in the cumulative runs. (c) Coefficient of variation (CV) values of quantification. Medians are indicated. Source data are provided as a Source Data file.

- The entrapment searches used to provide an empirical FDR for glycan composition assignments in the DIA analyses are too simple to provide a reasonable estimate of the composition-level FDR. In the fission yeast dataset, excluding glycan compositions with fucose and sialic acid is (or should be) trivial as there are no glycans with those monosaccharides in the yeast and thus no spectra should contain any specific fragment ions. The key challenge that needs to be assessed for glycopeptide DIA FDR is the ability of the method to correctly assign the glycan composition of a glycopeptide in the presence of glycan fragments from other glycopeptides in the chimeric DIA spectrum. If there are no NeuAc-specific oxonium ions in any glycan on any glycopeptide, for example, then not assigning any glycans with NeuAc is trivial and the method should appear to provide a perfect empirical FDR in the entrapment search. But in the human serum dataset, or any other dataset with NeuAc-containing glycans, correctly assigning the glycan compositions is not trivial as many spectra will have NeuAc-oxonium ions from some, but not necessarily all, glycopeptides in the spectrum. The accuracy of glycan compositions assigned in these cases is never assessed in this framework. The entrapment searches should be revised to include glycans with shared monosaccharides but nonsensical or not-present compositions to provide an empirical FDR assessment, and these limitations should be discussed in the text.

- o This is particularly important for assessing the FDR of the extended library. My assumption is that adding glycopeptides with glycans that contain NeuAc and/or Fuc but are not known in humans to an extended library would result in a substantially higher entrapment rate (though I would be happy to be proven wrong).

Response: Thank you for the comment. In the revision, we adopted entrapments with common monosaccharides but nonsensical compositions in addition to the original entrapments with unlikely monosaccharides. For each human glycan (with or without Fuc or NeuAc) in the glycan database used in the study, all of its branch Hex were replaced with HexNAc (while the core Hex were kept). Among the modified glycans, those with monosaccharide compositions not shared with the original human glycans were selected as entrapment glycans. These glycans (with 3 Hex and at least 8 HexNAc) are unlikely found in human. Entrapment entries were generated using glycopeptides with peptide sequences from human and the entrapment glycans, whereafter those with m/z out of DIA window range were removed. The composition-based entrapment benchmarking is stricter than the monosaccharide-based one. Notably, it cannot be used for DDA-based spectral libraries since there are not experimental spectra of these nonsensical glycopeptides.

Using the composition-based entrapments, the entrapment percentage using PredLib was 2.2%. For the extended libraries, PredExt 5k and 7k resulted in entrapment percentages of 3.4% and 2.3%, respectively. As the library coverage continued increasing (PredExt 10k), substantially more entrapment hits occurred, while the number of detected target glycopeptides plateaued, leading to a higher entrapment percentage (4.0%). Large libraries contain a significant fraction of “false target” glycopeptides not detectable in the samples, which would not boost the number of detected analytes. Instead, it is adverse to error rate control, which is currently the main

limitation of predicted spectral libraries.

The results are presented in **Supplementary Note 10**. Corresponding revision was made on page 15.

Supplementary Fig. 56. DIA results of using the entrapment libraries.

(a) Numbers of identifications from the fission yeast sample using the glycan entrapment libraries (containing glycopeptides with peptide sequences from yeast and glycans from human). (b) Results from the human serum sample using the monosaccharide-based entrapments. (c) Results from the human serum sample using the composition-based entrapments. The entrapment libraries are generated based on the experimental (DDALib) and the predicted spectral libraries (PredLib and PredExt), where library entries with glycans not present in the sample are appended to the original libraries with a size ratio of approximately 1:1. Results of DDALib are from the original publication of GproDIA.

Minor:

- The methods section states that the order of glycan branches is ignored, which is understandable given the limitations of glycan structure assignment from a single glycopeptide HCD spectrum. However, the branching order can have biological implications, so this caveat needs to be provided when describing the results as having determined the “correct structure” or the method as being “structure specific”.

Response: Thank you for the suggestions. In the revision, we clarified that the branching order was ignored when describing the results (on page 10).

- The text describing the DIA results claims that “in line with previous studies on conventional peptides and phosphopeptides...the library with predicted spectrum performed better”, however, the serum library with RT prediction was not better and neither of the yeast libraries with prediction performed better. In conventional proteomics and phosphoproteomics, retention time prediction typically provides a large performance boost, unlike the effect seen here. This statement should be revised to more accurately reflect the data presented.

Response: Thank you for the comment. In the revision, we modified this statement as follow (on page 14):

“ The library with predicted spectrum performed better than the experimental library. A possible reason is that predicted spectra can exceed the quality of some spectra in the original experimental library, e.g., those with incorrect identifications that did not accurately represent relative fragment ion intensities, as reported in previous studies on conventional proteomics and phosphoproteomics.

- Several of the Supplementary Data tables do not contain the protein ID for the identified GPSMs, making it hard to assess the results, particularly for the analysis of the glycoprotein standards. Please consider including this information to the tables.

Response: Thank you for the suggestions. In the revision, we added protein information to the Supplementary Data tables.

Reviewer #2 (Remarks to the Author):

The revised manuscript addressed most of our comments in the previous round of review. I have a few remaining concerns and a couple of suggestions as summarized below.

1. It is nice that the authors deposited their experimental MS data to ProteomeXchange and showed the project IDs of the public datasets used in the study. However, the GPSMs used for training and cross-validation (eg. for producing the data presented in Supplementary Note 3) were not made available, which makes it impossible to reproduce and evaluate the results, because specific procedure were adopted to pre-process the experimental MS/MS spectra into consensus spectra for the training/testing purpose. I strongly encourage the authors to release both the replicated and consensus spectra used for training/testing the model as supplementary data, allowing for replicating the results as well as follow-up studies.

2. The prediction achieved very high accuracy ($DP > 0.95$ in most cases). I wonder what is the coverage of the predicted peaks among the full experimental spectra, i.e., what is the fraction of peaks (and peak intensities) in the experimental spectra can be predicted based on the fragmentation considered in the study? What are the likely sources of the other peaks observed in the experimental spectra but not covered by the current prediction?

3. The author claim that "the order of glycan branches is hardly provided by the conventional HCD MS/MS-based glycopeptide identification method". This may be true, but it also means the replicated spectra of the "same glycopeptide" may result from isomeric glycopeptides. Can this lead to the variation among replicated spectra? This limitation should be discussed in the manuscript.

4. It appears the ratio of merging glycan/peptide fragments is a dataset-specific parameter (perhaps associated with the experimental condition?). How much was it influenced by peptide/glycan composition? I.e., is the variation of ratio contributed more by different datasets (due to experimental conditions) or more by different glycopeptides?

Reviewer #3 (Remarks to the Author):

I co-reviewed this manuscript with one of the reviewers who provided the listed reports. This is part of the Nature Communications initiative to facilitate training in peer review and to provide appropriate recognition for Early Career Researchers who co-review manuscripts

Reviewer #4 (Remarks to the Author):

The revised manuscript successfully addresses the issues present in the initial version. While there some remaining limitations with regards to prediction accuracy (understandable given the limitations of the available training data) and FDR control of extended spectral libraries, these are now clearly documented in the supplementary data and discussion.

Reviewer #4 (Remarks on code availability):

The included copy of the Github repository the authors say they will make public after the manuscript is published has clear documentation describing how to set up and run the code to reproduce the results of the paper. I have not reproduced the analysis myself given the very long training time required for the deep learning, but it appears to be sufficiently documented and described. There is not a user application, just Python scripts, so the method is limited to users who are comfortable working with Python code and configuring Python installations.

Responses to the Reviewer's Comments (Revision #2)

Prediction of glycopeptide fragment mass spectra by deep learning

Yi Yang et al.

Reviewer #2 (Remarks to the Author):

The revised manuscript addressed most of our comments in the previous round of review. I have a few remaining concerns and a couple of suggestions as summarized below.

Response: We acknowledge the reviewer for all valuable comments and suggestions, which helped us to improve the quality of the manuscript.

1. It is nice that the authors deposited their experimental MS data to ProteomeXchange and showed the project IDs of the public datasets used in the study. However, the GPSMs used for training and cross-validation (eg. for producing the data presented in Supplementary Note 3) were not made available, which makes it impossible to reproduce and evaluate the results, because specific procedure were adopted to pre-process the experimental MS/MS spectra into consensus spectra for the training/testing purpose. I strongly encourage the authors to release both the replicated and consensus spectra used for training/testing the model as supplementary data, allowing for replicating the results as well as follow-up studies.

Response: Thank you for the suggestions. We have deposited all the data for replicating the results, including both the replicated and consensus spectra that were used for training and cross-validation, to ProteomeXchange/iProX with accession number PXD045248/IPX0007075000. The data were organized as subprojects for different sections of the manuscript. The "FileAnnotation_*.xlsx" spreadsheet in each subproject shows the description of each file and its relationship with other files. The processed spectral data are named as "*.speclib.h5".

2. The prediction achieved very high accuracy ($DP > 0.95$ in most cases). I wonder what is the coverage of the predicted peaks among the full experimental spectra, i.e., what is the fraction of peaks (and peak intensities) in the experimental spectra can be predicted based on the fragmentation considered in the study? What are the likely sources of the other peaks observed in the experimental spectra but not covered by the current prediction?

Response: Thank you for the comment. We have added **Supplementary Fig. 31** showing the coverage and intensity fraction of peaks that were considered in this study among the raw experimental spectra. We have stated this information in the revised manuscript (on page 9):

“ Notably, similarity metrics were reported against the annotated b/y and B/Y ions accounting for 26–34% intensity of the raw experimental spectra (**Supplementary Fig. 31**). Other peaks, such as those sourced from noncanonical fragments or noise signals, were not covered in this study.

Supplementary Fig. 31. Coverage of the fragment peaks that were considered in this study among the raw experimental spectra.

(a) The coverage of peak numbers. (b) The fraction of the summed intensity of the covered peaks. The center lines indicate the median values. The lower/upper hinges of the boxes indicate the first/third quartiles, and the lower/upper whiskers extend from the hinges to the smallest/largest value no further than 1.5 times the interquartile range. The number of spectra (n) is indicated. Source data are provided as a Source Data file.

3. The author claim that "the order of glycan branches is hardly provided by the conventional HCD MS/MS-based glycopeptide identification method". This may be true, but it also means the replicated spectra of the "same glycopeptide" may result from isomeric glycopeptides. Can this lead to the variation among replicated spectra? This limitation should be discussed in the manuscript.

Response: Thank you for the comment. We agree that isomeric glycopeptides may be one of the factors contributing to the variation among replicated spectra. We have stated this limitation in the revised manuscript (on page 7):

“ Due to the limitation that our model ignores the order of glycan branches, isomeric glycopeptides may be one of the factors contributing to the variation among replicated spectra.

4. It appears the ratio of merging glycan/peptide fragments is a dataset-specific parameter (perhaps associated with the experimental condition?). How much was it influenced by peptide/glycan composition? Ie, is the variation of ratio contributed more by different datasets (due to experimental conditions) or more by different glycopeptides?

Response: Thank you for the comment. We agree that the ratio may be associated with the experimental condition. However, the variation of ratio was contributed more by different glycopeptides than by different datasets. We have stated this information in the revised manuscript (on page 8):

“ Within each dataset, the intensity ratio between peptide and glycan fragments varied in a wide range across different glycopeptides, while most glycopeptides had lower intensity ratios (**Supplementary Fig. 8**). Variations of the ratio were also observed across datasets, probably associated with the experimental condition.

Supplementary Fig. 8. Intensity ratio between peptide and glycan fragments calculated on consensus spectra.

(a) Distributions of intensity ratio values in the mouse and yeast datasets. Spectra prediction was performed using the model trained with Mouse 1. (b) Distributions for the human datasets. Spectra prediction was performed using the model trained with Human 1. The median values of the predicted (Pred) and experimental (Expr) intensity ratio are indicated. Source data are provided as a Source Data file.

Reviewer #3 (Remarks to the Author):

I co-reviewed this manuscript with one of the reviewers who provided the listed reports. This is part of the Nature Communications initiative to facilitate training in peer review and to provide appropriate recognition for Early Career Researchers who co-review manuscripts

Reviewer #4 (Remarks to the Author):

The revised manuscript successfully addresses the issues present in the initial version. While there are some remaining limitations with regards to prediction accuracy (understandable given the limitations of the available training data) and FDR control of extended spectral libraries, these are now clearly documented in the supplementary data and discussion.

Response: We acknowledge the reviewer for all valuable comments and suggestions, which helped us to improve the quality of the manuscript.

Reviewer #4 (Remarks on code availability):

The included copy of the Github repository the authors say they will make public after the manuscript is published has clear documentation describing how to set up and run the code to reproduce the results of the paper. I have not reproduced the analysis myself given the very long training time required for the deep learning, but it appears to be sufficiently documented and described. There is not a user application, just Python scripts, so the method is limited to users who are comfortable working with Python code and configuring Python installations.

Response: Thank you for the comment. We provided Python scripts due to the challenge of compiling a binary distribution for platforms and GPU models. Since the users may have different system environment, they can use the package manager to install the dependency libraries for a specific OS, GPU driver, and CUDA version. Users with programming experience can also integrate DeepGlyco in their informatics workflows.